# Preventing Conflicting Gradients in Neural Marked Temporal Point Processes

**Tanguy Bosser**                                                    *tanguy.bosser@umons.ac.be*
*Department of Computer Science*
*University of Mons*

**Souhaib Ben Taieb**                                               *souhaib.bentaieb@mbzuai.ac.ae*
*Mohamed bin Zayed University of Artificial Intelligence*
*University of Mons*

**Reviewed on OpenReview:** *https://openreview.net/forum?id=INijCSPtbQ*

## Abstract

Neural Marked Temporal Point Processes (MTPP) are flexible models to capture complex temporal inter-dependencies between labeled events. These models inherently learn two predictive distributions: one for the arrival times of events and another for the types of events, also known as marks. In this study, we demonstrate that learning a MTPP model can be framed as a two-task learning problem, where both tasks share a common set of trainable parameters that are optimized jointly. We show that this often leads to the emergence of conflicting gradients during training, where task-specific gradients are pointing in opposite directions. When such conflicts arise, following the average gradient can be detrimental to the learning of each individual tasks, resulting in overall degraded performance. To overcome this issue, we introduce novel parametrizations for neural MTPP models that allow for separate modeling and training of each task, effectively avoiding the problem of conflicting gradients. Through experiments on multiple real-world event sequence datasets, we demonstrate the benefits of our framework compared to the original model formulations.

## 1 Introduction

Sequences of labeled events observed in continuous time at irregular intervals are ubiquitous across various fields such as healthcare (Enguehard et al., 2020), finance (Bacry et al., 2015), social media (Farajtabar et al., 2017), and seismology (Ogata, 1998). In numerous application domains, an important problem involves predicting the timing and types of future events—often called marks—based on historical data. Marked Temporal Point Processes (MTPP) (Daley & Vere-Jones, 2003) provide a mathematical framework for modeling sequences of events, enabling subsequent inferences on the system under study. The Hawkes process, originally introduced by Hawkes (Hawkes, 1971), is a well-known example of a MTPP model that has found successful applications in diverse domains, including finance (Hawkes, 2018), crime analysis (Egesdal et al., 2010), and user recommendations (Du et al., 2015). However, the strong modeling assumptions of classical MTPP models often limit their ability to capture complex event dynamics (Mei & Eisner, 2017). This limitation has led to the rapid development of a more flexible class of neural MTPP models, incorporating recent advances in deep learning (Shchur et al., 2021).

This paper argues that learning a neural MTPP model can be interpreted as a two-task learning problem where both tasks share a common set of parameters and are optimized jointly. Specifically, one task focuses on learning the distribution of the next event's arrival time conditional on historical events. The other task involves learning the distribution of the categorical mark conditional on both the event's arrival time and the historical events. We identify these tasks as the time prediction and mark prediction, respectively. While parameter sharing between tasks can sometimes enhance training efficiency (Standley et al., 2020), it may

also result in performance degradation when compared to training each task separately. A major challenge in the simultaneous optimization of multi-task objectives is the issue of conflicting gradients (Liu et al., 2021b). This term describes situations where task-specific gradients point in opposite directions. When such conflicts arise, gradient updates tend to favor tasks with larger gradient magnitudes, thus hindering the learning process of other concurrent tasks and adversely affecting their performance. Although the phenomenon of conflicting gradients has been studied in various fields (Chen et al., 2018; 2020; Yu et al., 2020; Shi et al., 2023), its impact on the training of neural MTPP models remains unexplored. We propose to address this gap with the following contributions:

(1) We demonstrate that conflicting gradients frequently occur during the training of neural MTPP models. Furthermore, we show that such conflicts can significantly degrade a model's predictive performance on the time and mark prediction tasks.

(2) To prevent the issue of conflicting gradients, we introduce novel parametrizations for existing neural TPP models, allowing for separate modeling and training of the time and mark prediction tasks. Inspired from the success of (Shi et al., 2023), our framework allows to prevent gradient conflicts from the root while maintaining the flexibility of the original parametrizations.

(3) We want to emphasize that our approach to disjoint parametrizations does not assume the independence of arrival times and marks. Unlike prior studies that assumed conditional independence (Shchur et al., 2020; Du et al., 2016), we propose a simple yet effective parametrization for the mark conditional distribution that relaxes this assumption.

(4) Through a series of experiments with real-world event sequence datasets, we show the advantages of our framework over the original model formulations. Specifically, our framework effectively prevents the emergence of conflicting gradients during training, thereby enhancing the predictive accuracy of the models. Additionally, all our experiments are reproducible and implemented using a common code base[1].

## 2    Background and Notations

A **Marked Temporal Point Process** (MTPP) is a random process whose realization is a sequence of events $\mathcal{S} = \{e_i = (t_i, k_i)\}_{i=1}^n$. Each event $e_i \in \mathcal{S}$ is an ordered pair with an *arrival time* $t_i \in [0, T]$ (with $t_0 = 0$) and a categorical label $k_i \in \mathbb{K} = \{1, ..., K\}$ called *mark*. The arrival times form a sequence of strictly increasing random values observed within a specified time interval $[0, T]$, i.e. $0 \leq t_1 < t_2 < \ldots < t_n \leq T$. Equivalently, $\tau_i = t_i - t_{i-1} \in \mathbb{R}_+$ is an event *inter-arrival time*. We will use both representations interchangeably throughout the paper. If $e_{i-1}$ is the last observed event, the occurrence of the next event $e_i$ in $(t_{i-1}, \infty[$ can be fully characterized by the *joint PDF* $f(\tau, k | \mathcal{H}_t)$, where $\mathcal{H}_t = \{(t_i, k_i) \in \mathcal{S} \mid t_i < t\}$ is the observed process history. For clarity of notations, we will use the notation '∗' of (Daley & Vere-Jones, 2008) to indicate dependence on $\mathcal{H}_t$, i.e. $f^*(\tau, k) = f(\tau, k | \mathcal{H}_t)$. This joint PDF can be factorized as $f^*(\tau, k) = f^*(\tau) p^*(k | \tau)$, where $f^*(\tau)$ is the *PDF of inter-arrival times*, and $p^*(k | \tau)$ is the *conditional PMF of marks*. A MTPP can be equivalently described by its so-called *marked conditional intensity functions* (Rasmussen, 2018), which are defined for $t_{i-1} < t < t_i$ as $\lambda_k^*(t) = f^*(t, k) / (1 - F^*(t))$, where $F^*(t)$ is the *conditional CDF of arrival-times* and $k \in \mathbb{K}$. From the marked intensities, we can also define the *marked compensators* $\Lambda_k^*(t) = \int_{t_{i-1}}^t \lambda_k^*(s) ds$. Provided that certain modeling constraints are satisfied, each of $f^*(\tau, k)$, $\lambda_k^*(t)$ and $\Lambda_k^*(t)$ fully characterizes a MTPP and can be retrieved from the others (Rasmussen, 2018; Enguehard et al., 2020).

**Neural Temporal Point Processes.** To capture complex dependencies between events, the framework of neural MTPP (Shchur et al., 2021; Bosser & Ben Taieb, 2023) incorporates neural network components into the model architecture, allowing for more flexible models. A neural MTPP model consists of three main components: (1) An *event encoder* that learns a representation $\boldsymbol{e}_i \in \mathbb{R}^{d_e}$ for each event $e_i \in \mathcal{S}$, (2) a *history encoder* that learns a compact history embedding $\boldsymbol{h}_i \in \mathbb{R}^{d_h}$ of the history $\mathcal{H}_{t_i}$ of event $e_i$, and (3) a *decoder* that defines a function characterizing the MTPP from $\boldsymbol{h}_i$, e.g. $\lambda_k^*(t)$. Let $f^*(\tau, k; \boldsymbol{\theta})$, $\lambda_k^*(t; \boldsymbol{\theta})$, or $\Lambda_k^*(t; \boldsymbol{\theta})$ be a valid model of $f^*(\tau, k)$, where the set of trainable parameters $\boldsymbol{\theta}$ lies within the parameter space $\Theta$.

---

[1] https://github.com/tanguybosser/grapTPP_tmlr

To train this model, we use a dataset $\mathcal{S} = \{\mathcal{S}_1, ..., \mathcal{S}_L\}$, where each sequence $\mathcal{S}_l$ comprises $n_l$ events with arrival times observed within the interval $[0, T]$ and $l = 1, ..., L$. The training objective is the average sequence negative log-likelihood (NLL) (Rasmussen, 2018), given by

$$\mathcal{L}(\boldsymbol{\theta}; \mathcal{S}) = -\frac{1}{L} \sum_{l=1}^{L} \left[ \left[ \sum_{i=1}^{n_l} \log f^*(\tau_{l,i}, k_{l,i}; \boldsymbol{\theta}) \right] - \log(1 - F^*(T; \boldsymbol{\theta})) \right], \tag{1}$$

which is optimized using mini-batch stochastic gradient descent (Ruder, 2017).

## 3 Conflicting Gradients in Two-Task Learning for Neural MTPP Models

Consider the factorization of $f^*(\tau_{l,i}, k_{l,i}; \boldsymbol{\theta})$ into $f^*(\tau_{l,i}; \boldsymbol{\theta})$ and $p^*(k_{l,i}|\tau_{l,i}; \boldsymbol{\theta})$, where $f^*(\tau_{l,i}, k_{l,i}; \boldsymbol{\theta}) = f^*(\tau_{l,i}; \boldsymbol{\theta}) \cdot p^*(k_{l,i}|\tau_{l,i}; \boldsymbol{\theta})$. Substituting this decomposition into the NLL in equation 1 and rearranging terms, we obtain:

$$\mathcal{L}(\boldsymbol{\theta}; \mathcal{S}) = \underbrace{-\frac{1}{L} \sum_{l=1}^{L} \left[ \left[ \sum_{i=1}^{n_l} \log f^*(\tau_{l,i}; \boldsymbol{\theta}) \right] - \log(1 - F^*(T; \boldsymbol{\theta})) \right]}_{\mathcal{L}_T(\boldsymbol{\theta}, \mathcal{S})} \underbrace{-\frac{1}{L} \sum_{l=1}^{L} \left[ \sum_{i=1}^{n_l} \log p^*(k_{l,i}|\tau_{l,i}; \boldsymbol{\theta}) \right]}_{\mathcal{L}_M(\boldsymbol{\theta}, \mathcal{S})}. \tag{2}$$

This shows that the total objective function $\mathcal{L}(\boldsymbol{\theta}; \mathcal{S})$ consists of two sub-objectives: $\mathcal{L}_T(\boldsymbol{\theta}; \mathcal{S})$ and $\mathcal{L}_M(\boldsymbol{\theta}; \mathcal{S})$, revealing that learning an MTPP model can be interpreted as a *two-task* learning problem. The first objective, $\mathcal{L}_T(\boldsymbol{\theta}; \mathcal{S})$, relates to modeling the predictive distribution of (inter-)arrival times, which we refer to as the *time prediction task* $\mathcal{T}_T$. The second objective, $\mathcal{L}_M(\boldsymbol{\theta}; \mathcal{S})$, concerns modeling the conditional predictive distribution of the marks $p^*(k|\tau; \boldsymbol{\theta})$, which we call the *mark prediction task* $\mathcal{T}_M$.

**Conflicting gradients.** Assuming that $\mathcal{L}_T(\boldsymbol{\theta})$ and $\mathcal{L}_M(\boldsymbol{\theta})$ are differentiable, let $\boldsymbol{g}_T = \nabla_{\boldsymbol{\theta}} \mathcal{L}_T(\boldsymbol{\theta})$ and $\boldsymbol{g}_M = \nabla_{\boldsymbol{\theta}} \mathcal{L}_M(\boldsymbol{\theta})$ denote the gradients of $\mathcal{L}_T(\boldsymbol{\theta})$ and $\mathcal{L}_M(\boldsymbol{\theta})$, respectively, with respect to the shared parameters $\boldsymbol{\theta}$[2]. As discussed in (Shi et al., 2023), when $\boldsymbol{g}_T$ and $\boldsymbol{g}_M$ are pointing in opposite directions, i.e. $\boldsymbol{g}_T \cdot \boldsymbol{g}_M < 0$, an update step in the direction of negative $\boldsymbol{g}_T$ for $\boldsymbol{\theta}$ will increase the loss for task $\mathcal{T}_M$, and inversely for task $\mathcal{T}_T$ if an update step is taken in the direction of negative $\boldsymbol{g}_M$. Such *conflicting gradients* can be formally defined as follows.

**Definition 1** (Conflicting gradients (Shi et al., 2023)) *Let $\phi_{TM} \in [0, 2\pi]$ be the angle between the gradients $\boldsymbol{g}_T$ and $\boldsymbol{g}_M$. They are said to be conflicting with each other if $\cos \phi_{TM} < 0$.*

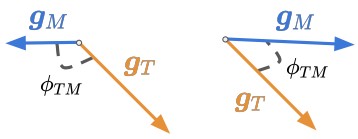

(a) Conflicting      (b) Non-conflicting

The smaller the value of $\cos \phi_{TM} \in [-1, 0]$, the more severe the conflict between the gradients. Figure (1) illustrates these conflicting gradients. Ideally, we want the gradients to align during optimization (i.e. $\cos \phi_{TM} > 0$) to encourage positive reinforcement between the two tasks, or to be simply orthogonal (i.e. $\cos \phi_{TM} = 0$). Conflicting gradients, especially those with significant differences in magnitude,

Figure 1: Conflicting gradients

pose substantial challenges during the optimization of multi-task learning objectives (Yu et al., 2020). Specifically, if $\boldsymbol{g}_T$ and $\boldsymbol{g}_M$ conflict, the update step for $\boldsymbol{\theta}$ will likely be dominated by the gradient of whichever task—$\boldsymbol{g}_T$ or $\boldsymbol{g}_M$—has the greater magnitude, thereby disadvantaging the other task. The degree of similarity between the magnitudes of these two gradients can be quantified using a metric known as *gradient magnitude similarity*, defined as follows:

**Definition 2** (Gradient Magnitude Similarity (Yu et al., 2020)) *The gradient magnitude similarity between $\boldsymbol{g}_T$ and $\boldsymbol{g}_M$ is defined as $GMS = \frac{2||\boldsymbol{g}_T||_2 ||\boldsymbol{g}_M||_2}{||\boldsymbol{g}_T||_2^2 + ||\boldsymbol{g}_M||_2^2}$, where $|| \cdot ||_2$ is the $l_2$-norm.*

A GMS value close to 1 indicates that the magnitudes of $\boldsymbol{g}_T$ and $\boldsymbol{g}_M$ are similar, while a GMS value close to 0 suggests a significant difference between them. Ideally, we aim to minimize the number of conflicting gradients and maintain a GMS close to 1 to ensure balanced learning across the two tasks. However, a

---

[2]We explicitly omit the dependency of $\mathcal{L}_T$ and $\mathcal{L}_M$ on $\mathcal{S}$ to simplify notations.

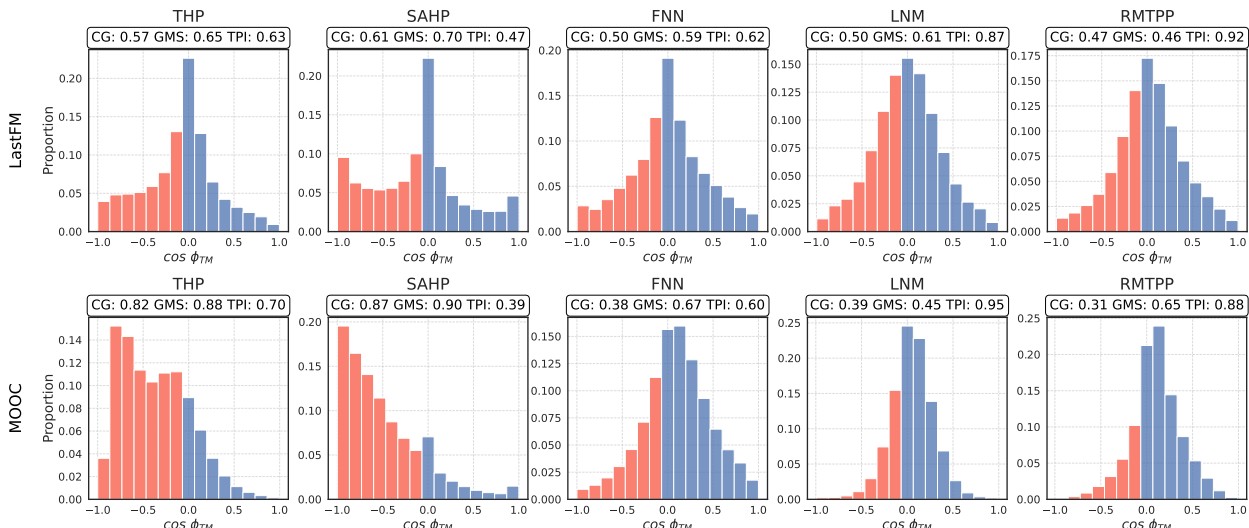

Figure 2: Distribution of cos $\phi_{TM}$ during training for the different baselines on MOOC and LastFM. CG refers to the proportion of cos $\phi_{TM} < 0$ observed during training. The distribution is obtained by pooling the values of $\phi_{TM}$ over 5 training runs, and gradients that are conflicting correspond to the red bars.

low GMS value among conflicting gradients does not specify which task is being prioritized. Therefore, we introduce the *time priority index* to address this issue, which is defined as follows:

**Definition 3** (Time Priority Index) *The time priority index between conflicting gradients $\boldsymbol{g}_T$ and $\boldsymbol{g}_M$ with* cos $\phi_{TM} < 0$ *is defined as* TPI $= \mathbb{1}\left(||\boldsymbol{g}_T||_2 > ||\boldsymbol{g}_M||_2\right)$ *where* $\mathbb{1}(\cdot)$ *is the indicator function.*

If $\boldsymbol{g}_T$ and $\boldsymbol{g}_M$ are conflicting and optimization prioritizes task $\mathcal{T}_T$, then the TPI takes the value 1. Conversely, if task $\mathcal{T}_M$ is prioritized, the TPI takes the value 0.

**Do gradients conflict in neural MTPP models?** To explore this question, we perform a preliminary experiment with common neural MTPP baselines that either learn $f^*(t, k; \boldsymbol{\theta})$, $\lambda_k^*(t; \boldsymbol{\theta})$, or $\Lambda_k^*(t; \boldsymbol{\theta})$: THP (Zuo et al., 2020), SAHP (Zhang et al., 2020), FNN (Omi et al., 2019), LNM (Shchur et al., 2020), and RMTPP (Du et al., 2016). We aim to determine whether conflicting gradients occur during their training. For this purpose, each model is trained to minimize the NLL defined in (2) using sequences from two real-world datasets: LastFM (Hidasi & Tikk, 2012) and MOOC (Kumar et al., 2019). For optimization, we rely on the Adam optimizer (Kingma & Ba, 2014) used by default in neural MTPP training with learning rate $\alpha = 10^{-3}$. At every gradient update, we calculate the gradients $\boldsymbol{g}_T$ and $\boldsymbol{g}_M$ with respect to the shared parameters $\boldsymbol{\theta}^3$, recording values of cos $\phi_{TM}$, GMS, and TPI. Figure 2 shows the distribution of cos $\phi_{TM}$ across all $S$ training iterations, along with the average values of GMS and TPI, and the proportion of conflicting gradients (CG), computed as

$$\text{CG} = \frac{\sum_{s=1}^S \mathbb{1}\left(\cos \phi_{TM}^s < 0\right)}{\sum_{s=1}^S \cos \phi_{TM}^s}, \tag{3}$$

where $\phi_{TM}^s$ refers to the angle between $\boldsymbol{g}_T^s$ and $\boldsymbol{g}_M^s$ at training iteration $s$. Gradients that are conflicting during training correspond to the red bars. We observe that some models, such as THP and SAHP on MOOC, frequently exhibit conflicting gradients during training, as indicated by a high value of CG. Conversely, while other models show a more balanced proportion of conflicting gradients, these are generally characterized by low GMS values, which may potentially impair performance on the task with the lowest magnitude gradient. In this context, the data indicates that optimization generally tends to favor $\mathcal{T}_T$ during optimization, as suggested by an average TPI greater than 0.5. In Section 6, our experiments show that the combined influence of a high proportion of conflicts and low GMS values during training can significantly deteriorate model performance on the time and mark prediction tasks. We provide additional visualizations for other datasets in Appendix H.5, which confirm the above discussion and further suggest that conflicting gradients emerge frequently during the training of neural MTPP models.

---

[3]In practice, conflicts are computed layer-wise for each layer of the model (e.g the weights $\mathbf{W}$ of a fully-connected layer).

## 4  A Framework to Prevent Conflicting Gradients in Neural MTPP Models

Given the observations from the previous section, our goal is to prevent the occurrence of conflicting gradients during the training of neural MTPP models with the NLL objective given in (2). To accomplish this, we first propose in Section 4.1 a naive approach that leverages duplicated and disjoint instances of the same model. Then, to avoid redundancy in model specification, we introduce in Section 4.2 novel parametrizations for neural MTPP models. We finally show in Section 4.3 how our parametrizations enable disjoint modeling and training of the time and mark prediction tasks.

Existing MTPP models generally fall into three categories based on the chosen parametrization: (1) *Intensity-based* approaches that model the marked intensity function $\lambda_k^*(t; \boldsymbol{\theta})$ (Zuo et al., 2020), (2) *Density-based* approaches that model the joint density $f^*(t, k; \boldsymbol{\theta})$ (Shchur et al., 2020), and (3) *Compensator-based* approaches that model the marked compensators $\Lambda_k^*(t; \boldsymbol{\theta})$ (Omi et al., 2019). We denote these different approaches as *joint parametrizations* because they define a function that involves both the arrival time and the mark. To enable disjoint modeling of the time and mark prediction tasks, we consider the factorization of these functions into the products of two components: one involving a function of the arrival-times, and the other involving the conditional PMF of marks:

$$\lambda_k^*(t; \boldsymbol{\theta}) = \lambda^*(t; \boldsymbol{\theta})p^*(k|t; \boldsymbol{\theta}) = \frac{d}{dt}\left[\Lambda^*(t; \boldsymbol{\theta})\right]p^*(k|t; \boldsymbol{\theta}), \tag{4}$$

$$f^*(t, k; \boldsymbol{\theta}) = f^*(t; \boldsymbol{\theta})p^*(k|t; \boldsymbol{\theta}), \tag{5}$$

$$\Lambda_k^*(t; \boldsymbol{\theta}) = \int_{t_{i-1}}^{t} \lambda^*(s; \boldsymbol{\theta})p^*(k|s; \boldsymbol{\theta})ds, \tag{6}$$

where $\lambda^*(t; \boldsymbol{\theta}) = \frac{f^*(t; \boldsymbol{\theta})}{1-F^*(t; \boldsymbol{\theta})}$ and $\Lambda^*(t; \boldsymbol{\theta}) = \int_{t_{i-1}}^{t} \lambda^*(s; \boldsymbol{\theta})ds$ are respectively the *ground intensity* and the *ground compensator* of the process. Similarly to (2), a two-task NLL objective involving the r.h.s of expressions (4) and (6) can be derived, where task $\mathcal{T}_T$ now consists in learning either $\lambda^*(t; \boldsymbol{\theta})$ or $\Lambda^*(t; \boldsymbol{\theta})$. We provide the expression of these two-task NLL objectives in Appendix (A). The factorizations presented in expressions (4) and (5) show that, to enable disjoint modeling of time and mark prediction functions, we need to specify $p^*(k|t; \boldsymbol{\theta}_M)$ with parameters $\boldsymbol{\theta}_M$, and either $f^*(t; \boldsymbol{\theta}_T)$, $\lambda^*(t; \boldsymbol{\theta}_T)$, or $\Lambda^*(t; \boldsymbol{\theta}_T)$ with parameters $\boldsymbol{\theta}_T$. As a naive approach, we first explore obtaining these functions from duplicated instances of the joint parametrizations $\lambda_k^*(t; \boldsymbol{\theta})$, $\Lambda_k^*(t; \boldsymbol{\theta})$ or $f^*(t, k; \boldsymbol{\theta})$, as presented next.

### 4.1  A Naive Approach to Achieve Disjoint Parametrizations

Consider a model that parametrizes $\lambda_k^*(t; \boldsymbol{\theta})$, such as THP (Zuo et al., 2020). To obtain a disjoint parametrization of $\lambda^*(t; \boldsymbol{\theta}_T)$ and $p^*(k|t; \boldsymbol{\theta}_M)$, we can parametrize two identical functions $\lambda_k^*(t; \boldsymbol{\theta}_T)$ and $\lambda_k^*(t; \boldsymbol{\theta}_M)$ from the same model and for all $k \in \mathbb{K}$, where $\boldsymbol{\theta}_T$ and $\boldsymbol{\theta}_M$ are disjoint set of trainable parameters. From these two functions, we can finally derive $\lambda^*(t; \boldsymbol{\theta}_T)$ and $p^*(k|t; \boldsymbol{\theta}_M)$ as

$$\lambda^*(t; \boldsymbol{\theta}_T) = \sum_{k=1}^{K} \lambda_k^*(t; \boldsymbol{\theta}_T) \quad \text{and} \quad p^*(k|t; \boldsymbol{\theta}_M) = \frac{\lambda_k^*(t; \boldsymbol{\theta}_M)}{\sum_{k=1}^{K} \lambda_k^*(t; \boldsymbol{\theta}_M)}, \tag{7}$$

effectively defining the desired disjoint parametrization. In the presence of conflicts during training, we can show that a gradient update step for the shared model $\lambda_k^*(t; \boldsymbol{\theta})$ leads to higher loss compared to the duplicated model in (7) with disjoint parameters $\boldsymbol{\theta}_T$ and $\boldsymbol{\theta}_M$. Indeed, suppose that $\boldsymbol{\theta}$, $\boldsymbol{\theta}_T$ and $\boldsymbol{\theta}_M$ are all initialized with the same $\boldsymbol{\theta}^s$ at training iteration $s \in \mathbb{N}$. Assuming that $\mathcal{L}_T$ and $\mathcal{L}_M$ are differentiable, let

$$\boldsymbol{g}_T = \nabla_{\boldsymbol{\theta}} \mathcal{L}_T(\boldsymbol{\theta}^s) \quad \text{and} \quad \boldsymbol{g}_M = \nabla_{\boldsymbol{\theta}} \mathcal{L}_M(\boldsymbol{\theta}^s). \tag{8}$$

Denoting $\phi_{TM}$ as the angle between $\boldsymbol{g}_T$ and $\boldsymbol{g}_M$, we have the following corollary of Theorem 4.1. from (Shi et al., 2023):

**Corollary 1.** *Assume that $\mathcal{L}_T$ and $\mathcal{L}_M$ are differentiable, and that the learning rate $\alpha$ is sufficiently small. If $\cos \phi_{TM} < 0$, then $\mathcal{L}(\{\boldsymbol{\theta}_T^{s+1}, \boldsymbol{\theta}_M^{s+1}\}) < \mathcal{L}(\boldsymbol{\theta}^{s+1})$.*

This result essentially indicates that a model trained with disjoint parameters leads to lower loss after a gradient update if conflicts arise during training, i.e. $\cos \phi_{TM} < 0$. We provide the proof in Appendix C. Naturally, expression (7) and Corollary 1 remain valid for models that parametrize $\Lambda_k^*(t; \boldsymbol{\theta})$ or $f^*(\tau, k; \boldsymbol{\theta})$, as these functions can be uniquely retrieved from $\lambda_k^*(t; \boldsymbol{\theta})$.

### 4.2 Novel Disjoint Parametrizations of Neural MTPP Models

In this section, we introduce an alternative approach to (7) to achieve disjoint parametrizations of $\mathcal{T}_T$ and $\mathcal{T}_M$. Specifically, we introduce novel parametrizations of existing neural MTPP model that directly parametrize $p^*(k|t; \boldsymbol{\theta}_M)$ and either $f^*(t; \boldsymbol{\theta}_T)$, $\lambda^*(t; \boldsymbol{\theta}_T)$, or $\Lambda^*(t; \boldsymbol{\theta}_T)$, thereby avoiding the unnecessary redundancy in model specification required by the method in (7). For a query time $t \geq t_{i-1}$, we define a history representation $\boldsymbol{h} = \text{ENC}(\{\boldsymbol{e}_1, ..., \boldsymbol{e}_{i-1}\}; \boldsymbol{\theta}_h) \in \mathbb{R}^{d_h}$, where $\text{ENC}(\cdot \, ; \boldsymbol{\theta}_h)$ denotes the history encoder with parameters $\boldsymbol{\theta}_h$. The encoder $\text{ENC}(\cdot \, ; \boldsymbol{\theta}_h)$ is general and encompasses any encoder architecture typically found in the neural MTPP literature, such as recurrent neural networks (RNN) (Du et al., 2016; Shchur et al., 2020) or self-attention mechanisms (Zuo et al., 2020; Zhang et al., 2020).

**A general approach to model the distribution of marks.** Given a query time $t \geq t_{i-1}$ and its corresponding history representation $\boldsymbol{h}$, we propose to define the conditional PMF of marks $p^*(k|t; \boldsymbol{\theta}_M)$ using the following simple model:

$$p^*(k|t; \boldsymbol{\theta}_M) = \sigma_{\text{so}} \left( \mathbf{W}_2 \sigma_R \left( \mathbf{W}_1 \left[ \boldsymbol{h} || \log(\tau) \right] + \boldsymbol{b}_1 \right) + \boldsymbol{b}_2 \right), \tag{9}$$

where $\tau = t - t_{i-1}$, $\sigma_{\text{so}}$ is the softmax activation function, $\mathbf{W}_1 \in \mathbb{R}^{d_1 \times (d_h+1)}$, $\boldsymbol{b}_1 \in \mathbb{R}^{d_1}$, $\mathbf{W}_2 \in \mathbb{R}^{K \times d_1}$, $\boldsymbol{b}_2 \in \mathbb{R}^K$, and $||$ means concatenation. Here, $\boldsymbol{\theta}_M = \{\mathbf{W}_1, \mathbf{W}_2, \boldsymbol{b}_1, \boldsymbol{b}_2, \boldsymbol{\theta}_h\}$. Despite its simplicity, this model is flexible and capable of capturing the evolving dynamics of the mark distribution between two events. Note that equation 9 effectively captures inter-dependencies between arrival times and marks. Moreover, by removing $\log(t - t_{i-1})$ from expression (9), we obtain a PMF of marks that is independent of time, given $\mathcal{H}_t$.

**Intensity-based parametrizations.** We first consider MTPP models specified by their marked intensities, namely THP (Zuo et al., 2020) and SAHP (Zhang et al., 2020). We propose revising the original model formulations to directly parametrize $\lambda^*(t)$, while $p^*(k|t)$ is systematically derived from expression (9). Furthermore, although RMTPP (Du et al., 2016) is originally defined in terms of this decomposition, we extend the model to incorporate the dependence of marks on time.

$SAHP^+$. While the original model formulation parametrizes $\lambda_k^*(t)$, we adapt its formulation to define:

$$\lambda^*(t; \boldsymbol{\theta}_T) = \mathbf{1}^{\mathsf{T}} \left[ \sigma_S \left( \boldsymbol{\mu} - (\boldsymbol{\eta} - \boldsymbol{\mu}) \exp(-\boldsymbol{\gamma}(t - t_{i-1})) \right) \right], \tag{10}$$

where $\mathbf{1} \in \mathbb{R}^C$ is a vector of 1's allowing to define the ground intensity as a sum over $C$ different representations. In (10), $\boldsymbol{\mu} = \sigma_G(\mathbf{W}_\mu \boldsymbol{h})$, $\boldsymbol{\eta} = \sigma_S(\mathbf{W}_\eta \boldsymbol{h})$ and $\boldsymbol{\gamma} = \sigma_G(\mathbf{W}_\gamma \boldsymbol{h})$ where $\sigma_S$ and $\sigma_G$ are respectively the softplus and GeLU activation functions (Hendrycks & Gimpel, 2023). $\mathbf{W}_\mu, \mathbf{W}_\eta, \mathbf{W}_\gamma \in \mathbb{R}^{C \times d_h}$ are learnable parameters and $\boldsymbol{\theta}_T = \{\mathbf{W}_\mu, \mathbf{W}_\eta, \mathbf{W}_\gamma, \boldsymbol{\theta}_h\}$.

$THP^+$. Following a similar reasoning, the original formulation of THP is adapted to model $\lambda^*(t)$ instead of $\lambda_k^*(t)$:

$$\lambda^*(t; \boldsymbol{\theta}_T) = \mathbf{1}^{\mathsf{T}} \left[ \sigma_S \left( \boldsymbol{w}_t \frac{t - t_{i-1}}{t_{i-1}} + \mathbf{W} \boldsymbol{h} + \boldsymbol{b} \right) \right], \tag{11}$$

where $\boldsymbol{w}_t \in \mathbb{R}_+^C$, $\mathbf{W} \in \mathbb{R}^{C \times d_h}$, $\boldsymbol{b} \in \mathbb{R}^{d_1}$, and $\boldsymbol{\theta}_T = \{\mathbf{W}, \boldsymbol{w}_t, \boldsymbol{b}, \boldsymbol{\theta}_h\}$.

$RMTPP+$. We retain the original definition of $\lambda^*(t)$ proposed in RMTPP:

$$\lambda^*(t; \boldsymbol{\theta}_T) = \exp \left( w_t(t - t_{i-1}) + \boldsymbol{w}_h^{\mathsf{T}} \boldsymbol{h} + b \right), \tag{12}$$

where $w_t \in \mathbb{R}_+$, $\boldsymbol{w}_h \in \mathbb{R}^{d_h}$, $b \in \mathbb{R}$, and $\boldsymbol{\theta}_T = \{w_t, \boldsymbol{w}_h, b, \boldsymbol{\theta}_h\}$. A major difference between the original formulation of RMTPP and our approach lies in $p^*(k|\tau; \boldsymbol{\theta}_M)$ being defined by (9), which alleviates the original model assumption of marks being independent of time given $\mathcal{H}_t$.

**Density-based parametrizations.** The decomposition of the joint density in expression (5) has previously been considered by (Shchur et al., 2020) with the LogNormMix (LNM) model. However, similar to RMTPP, this model assumes independence of marks on time, given $\mathcal{H}_t$. In $LNM^+$, we relax this assumption by using expression (9) to parametrize $p^*(k|t)$, while maintaining a mixture of log-normal distributions for $f^*(\tau)$:

$$f^*(\tau; \boldsymbol{\theta}_T) = \sum_{m=1}^{M} p^*(m) \frac{1}{\tau \sigma_m \sqrt{2\pi}} \exp\left( -\frac{(\log \tau - \mu_m)^2}{2\sigma_m^2} \right), \tag{13}$$

where $\mu_m = (\mathbf{W}_\mu \boldsymbol{h} + \boldsymbol{b}_\mu)_m$, $p^*(m) = \sigma_{\text{so}}(\mathbf{W}_p \boldsymbol{h} + \boldsymbol{b}_p)_m$, $\sigma_m = \exp(\mathbf{W}_\sigma \boldsymbol{h} + \boldsymbol{b}_\sigma)_m$ with $\mathbf{W}_p, \mathbf{W}_\mu, \mathbf{W}_\sigma \in \mathbb{R}^{M \times d_h}$ and $\boldsymbol{b}_p, \boldsymbol{b}_\mu, \boldsymbol{b}_\sigma \in \mathbb{R}^M$, $M$ being the number of mixture components. Here $\boldsymbol{\theta}_T = \{\mathbf{W}_\mu, \mathbf{W}_p, \mathbf{W}_\sigma, \boldsymbol{b}_\mu, \boldsymbol{b}_p, \boldsymbol{b}_\sigma, \boldsymbol{\theta}_h\}$. Again, parametrizing the mark distribution using (9) relaxes the original assumption of marks being independent of time given $\mathcal{H}_t$.

**Compensator-based parametrizations.** Integrating compensator-based neural MTPP models into our framework requires to define $\Lambda^*(t; \boldsymbol{\theta}_T)$, this way retrieving the decomposition in the r.h.s. of (4). Specifically, we extend the improved marked FullyNN model (FNN) (Omi et al., 2019; Enguehard et al., 2020) into $FNN^+$, that models $\Lambda^*(t)$ and $p^*(k|t)$, instead of $\Lambda_k^*(t)$:

$$\Lambda^*(t; \boldsymbol{\theta}_T) = G^*(t) - G^*(t_{i-1}), \tag{14}$$

$$G^*(t) = \mathbf{1}^\mathsf{T} \left[ \sigma_S\big(\mathbf{W}(\sigma_{GS}(\boldsymbol{w}_t(t - t_{i-1}) + \mathbf{W}_h \boldsymbol{h} + \boldsymbol{b}_1) + \boldsymbol{b}_2)\big) \right], \tag{15}$$

where $\mathbf{W} \in \mathbb{R}^{C \times d_1}$, $\boldsymbol{w}_t, \boldsymbol{b}_1 \in \mathbb{R}^{d_1}$, $\mathbf{W}_h \in \mathbb{R}^{d_1 \times d_h}$, $\boldsymbol{b}_2 \in \mathbb{R}^C$, and $\sigma_{GS}$ is the Gumbel-softplus activation function. Here, $\boldsymbol{\theta}_T = \{\mathbf{W}, \mathbf{W}_h, \boldsymbol{w}_t, \boldsymbol{b}_1, \boldsymbol{b}_2, \boldsymbol{\theta}_h\}$. Similarly to the previous models, we use (9) to define $p^*(k|t)$.

**Training different history encoders**. The different functions defined in this section, $p^*(k|\tau; \boldsymbol{\theta}_M)$, $f^*(\tau; \boldsymbol{\theta}_T)$, $\lambda^*(t; \boldsymbol{\theta}_T)$, and $\Lambda^*(t; \boldsymbol{\theta}_T)$, share a common set of parameters $\boldsymbol{\theta}_h$ through a common history representation $\boldsymbol{h}$. To enable fully disjoint modeling and training of the time and mark predictive functions, we define two distinct history representations:

$$\boldsymbol{h}_T = \text{ENC}_T\left[\{\boldsymbol{e}_1, ..., \boldsymbol{e}_{i-1}\}; \boldsymbol{\theta}_{T,h}\right] \in \mathbb{R}^{d_h^t} \quad \text{and} \quad \boldsymbol{h}_M = \text{ENC}_M\left[\{\boldsymbol{e}_1, ..., \boldsymbol{e}_{i-1}\}; \boldsymbol{\theta}_{M,h}\right] \in \mathbb{R}^{d_h^m}, \tag{16}$$

where $\text{ENC}_T(\cdot\;; \boldsymbol{\theta}_{T,h})$ and $\text{ENC}_M(\cdot\;; \boldsymbol{\theta}_{M,h})$ are the *time* and *mark* history encoders, respectively, while $\boldsymbol{\theta}_{T,h}$ and $\boldsymbol{\theta}_{M,h}$ represent the sets of *disjoint* learnable parameters of the two encoders. By using $\boldsymbol{h}_T$ for $f^*(\tau; \boldsymbol{\theta}_T)$, $\lambda^*(t; \boldsymbol{\theta}_T)$ and $\Lambda^*(t; \boldsymbol{\theta}_T)$, and $\boldsymbol{h}_M$ for $p^*(k|\tau; \boldsymbol{\theta}_M)$[4], we have defined completely disjoint parametrizations of the decompositions in (4) and (5). Using separate history encoders further enables the model to capture information from past event occurrences that are relevant to the time and mark prediction tasks separately. In this paper, without loss of generality, we compute $\boldsymbol{h}_T$ and $\boldsymbol{h}_M$ by training two GRU encoders that sequentially process the set of event representations in $\{\boldsymbol{e}_1, ..., \boldsymbol{e}_{i-1}\}$.

### 4.3 Disjoint training of the time and mark tasks.

Let us model $f^*(\tau; \boldsymbol{\theta}_T)$ from (13) and $p^*(k|\tau; \boldsymbol{\theta}_M)$ from (9), using distinct history encoders such that $\boldsymbol{\theta}_T$ and $\boldsymbol{\theta}_M$ are disjoint set of trainable parameters. By injecting these expressions in (2), we find that the NLL now is a sum over two disjoint objectives $\mathcal{L}_T(\boldsymbol{\theta}_T, \mathcal{S})$ and $\mathcal{L}_M(\boldsymbol{\theta}_M, \mathcal{S})$, i.e.

$$\mathcal{L}(\boldsymbol{\theta}_T, \boldsymbol{\theta}_M; \mathcal{S}) = \underbrace{-\frac{1}{L} \sum_{l=1}^{L} \left[ \left[ \sum_{i=1}^{n_l} \log f^*(\tau_{l,i}; \boldsymbol{\theta}_T) \right] - \log(1 - F^*(T; \boldsymbol{\theta}_T)) \right]}_{\mathcal{L}_T(\boldsymbol{\theta}_T, \mathcal{S})} \underbrace{-\frac{1}{L} \sum_{l=1}^{L} \left[ \sum_{i=1}^{n_l} \log p^*(k_{l,i}|\tau_{l,i}; \boldsymbol{\theta}_M) \right]}_{\mathcal{L}_M(\boldsymbol{\theta}_M, \mathcal{S})}, \tag{17}$$

meaning that the associated tasks $\mathcal{T}_T$ and $\mathcal{T}_M$ can be learned separately. This contrasts with previous works, such as (Shchur et al., 2020) and (Zuo et al., 2020), in which shared parameters between $f^*$ and $p^*$ does not allow for disjoint training of (2). In our implementation, we minimize expression (17) through a single

---

[4]$\boldsymbol{\theta}_h$ is now replaced by $\boldsymbol{\theta}_{T,h}$ in $\boldsymbol{\theta}_T$, and by $\boldsymbol{\theta}_{M,h}$ in $\boldsymbol{\theta}_M$.

pipeline by specifying different early-stopping criteria for $\mathcal{L}_T(\boldsymbol{\theta}_T, \mathcal{S})$ and $\mathcal{L}_M(\boldsymbol{\theta}_M, \mathcal{S})$. Note that a similar decomposition can be obtained from any of the parametrizations presented in Sections 4.1 and 4.2. Finally, we would like to emphasize that disjoint training of tasks $\mathcal{T}_T$ and $\mathcal{T}_M$ through (17) does **not** imply independence of arrival-times and marks given the history. In fact, in our parametrizations, this dependency remains systematically captured by (9).

## 5 Related Work

**Neural MTPP models.** To address the limitations of simple parametric MTPP models (Hawkes, 1971; Isham & Westcott, 1979), prior studies focused on designing more flexible approaches by leveraging recent advances in deep learning. Based on the parametrization chosen, these neural MTPP models can be generally classified along three main axis: density-based, intensity-based, and compensator-based. Intensity-based approaches propose to model the trajectories of future arrival-times and marks by parametrizing the marked intensities $\lambda_k^*(t)$. In this line of work, past event occurrences are usually encoded into a history representation using RNNs (Du et al., 2016; Mei & Eisner, 2017; Guo et al., 2018b; Türkmen et al., 2019; Biloš et al., 2019; Zhu et al., 2020) or self-attention (SA) mechanisms (Zuo et al., 2020; Zhang et al., 2020; Zhu et al., 2021; Yang et al., 2022; Li et al., 2023). However, parametrizations of the marked intensity functions often come at the cost of being unable to evaluate the log-likelihood in closed-form, requiring Monte Carlo integration. This consideration motivated the design of compensator-based approaches that parametrize $\Lambda_k^*(t)$ using fully-connected neural networks Omi et al. (2019), or SA mechanisms (Enguehard et al., 2020), from which $\lambda_k^*(t)$ can be retrieved through differentiation. Finally, density-based approaches aim at directly modeling the joint density of (inter-)arrival times and marks $f^*(\tau, k)$. Among these, different family of distributions have been considered to model the distribution of inter-arrival times (Xiao et al., 2017; Lin et al., 2021). Notably Shchur et al. (2020) relies on a mixture of log-normal distributions (Shchur et al., 2020) to estimate $f^*(\tau)$, a model that then appeared in subsequent works (Sharma et al., 2021; Gupta et al., 2021). However, the original work of Shchur et al. (2020) assumes conditional independence of inter-arrival times and marks given the history, which is alleviated in (Waghmare et al., 2022). Nonetheless, a common thread of these parametrizations is that they explicitly enforce parameter sharing between the time and mark prediction tasks. As we have shown, this often leads to the emergence of conflicting gradient during training, potentially hindering model performance. For an overview of neural MTPP models, we refer the reader to the works of (Shchur et al., 2021), (Lin et al., 2022) and (Bosser & Ben Taieb, 2023).

**Conflicting gradients in multi-task learning.** Diverse approaches have been investigated in the literature to improve interactions between concurrent tasks in multi-task learning problems, thereby boosting performance for each task individually. In this context, a prominent line of work, called *gradient surgery methods*, focuses on balancing the different tasks at hand through direct manipulation of their gradients. These manipulations either aim at alleviating the differences in gradient magnitudes between tasks (Chen et al., 2018; Sener & Koltun, 2018; Liu et al., 2021c), or the emergence of conflicts (Sinha et al., 2018; Maninis et al., 2019; Yu et al., 2020; Chen et al., 2020; Wang et al., 2020; Liu et al., 2021a; Javaloy & Valera, 2022). Alternative approaches to task balancing have been explored based on different criteria, such as task prioritization (Guo et al., 2018a), uncertainty (Kendall et al., 2018), or learning pace (Liu et al., 2019). Our methodology relates more to branched architecture search approaches (Guo et al., 2020; Bruggemann et al., 2020; Shi et al., 2023), where the aim is set on dynamically identifying which layers should or should not be shared between tasks based on a chosen criterion, e.g. the proportion of conflicting gradients. Specifically, Shi et al. (2023) recently showed that gradient surgery approaches for multi-task learning objectives, such as GradDrop (Chen et al., 2020), PCGrad (Yu et al., 2020), CAGrad (Liu et al., 2021a), and MGDA (Sener & Koltun, 2018), cannot effectively reduce the occurrence of conflicting gradients during training. Instead, they propose to address task conflicts directly from the root by turning shared layer into task-specific layers if they experience conflicting gradients too frequently. Inspired by their success, our framework follows a similar approach: the time and mark prediction tasks are parametrized on disjoint set of trainable parameters to avoid conflicts from the root during training. We want to emphasize that our goal is not to propose a general-purpose gradient surgery method to mitigate the negative impact of conflicting gradients. Instead, we want to demonstrate that conflicts can be avoided altogether during the training of neural MTPP models by adapting their original parametrizations.

(a) Base                       (b) Base+                       (c) Base++

$\mathcal{H}_t \to \boxed{\text{ENC}(\cdot;\boldsymbol{\theta}_h)} \to \boxed{\text{DEC}(\cdot;\boldsymbol{\theta})} \to \lambda_k^*(t;\boldsymbol{\theta},\boldsymbol{\theta}_h)$

$\mathcal{H}_t \to \boxed{\text{ENC}(\cdot;\boldsymbol{\theta}_h)} \Big\langle \begin{array}{l} \boxed{\text{DEC}(\cdot;\boldsymbol{\theta}_T)} \to \lambda^*(t;\boldsymbol{\theta}_T,\boldsymbol{\theta}_h) \\ \boxed{\text{DEC}(\cdot;\boldsymbol{\theta}_M)} \to p^*(k|t;\boldsymbol{\theta}_M,\boldsymbol{\theta}_h) \end{array}$

$\mathcal{H}_t \Big\langle \begin{array}{l} \boxed{\text{ENC}(\cdot;\boldsymbol{\theta}_{T,h})} \to \boxed{\text{DEC}(\cdot;\boldsymbol{\theta}_T)} \to \lambda^*(t;\boldsymbol{\theta}_T,\boldsymbol{\theta}_{T,h}) \\ \boxed{\text{ENC}(\cdot;\boldsymbol{\theta}_{M,h})} \to \boxed{\text{DEC}(\cdot;\boldsymbol{\theta}_M)} \to p^*(k|t;\boldsymbol{\theta}_M,\boldsymbol{\theta}_{M,h}) \end{array}$

Figure 3: Graphical representation of the base, "+", and "++" setups.

# 6 Experiments

**Datasets and baselines.** We conduct an experimental study to assess the performance of our framework in training the time and mark prediction tasks from datasets composed of multiple event sequences. Specifically, we explore the various novel neural TPP parametrizations enabled by our framework, as detailed in Section 4. These are compared to their original parametrizations[5]. We use five real-world marked event sequence datasets frequently referenced in the neural MTPP literature: **LastFM** (Hidasi & Tikk, 2012), **MOOC**, **Reddit** (Kumar et al., 2019), **Github** (Trivedi et al., 2019), and **Stack Overflow** (Du et al., 2016). We provide descriptions and summary statistics for all datasets in Appendix D. We consider as base models **THP** (Zuo et al., 2020), **SAHP** (Zhang et al., 2020), **RMTPP** (Du et al., 2016), FullyNN (**FNN**) (Omi et al., 2019), LogNormMix (**LNM**) (Shchur et al., 2020), and SMURF-THP (**STHP**). To highlight the different components of our framework that lead to performance gains, we introduce the following settings:

(1) *Shared History Encoders and Disjoint Decoders.* A common history embedding, denoted as $\boldsymbol{h}$ is used, while the two functional terms from equations (4) and (5) are modeled separately as detailed in Section 4.2. Here, the functions $f^*(\tau;\boldsymbol{\theta}_T)$, $\lambda^*(t;\boldsymbol{\theta}_T)$, $\Lambda^*(t;\boldsymbol{\theta}_T)$ and $p^*(k|\tau;\boldsymbol{\theta}_M)$ share common parameters via $\boldsymbol{h}$. Models trained in this setting are indicated with a "+" sign, e.g. THP+.

(2) *Disjoint History Encoders and Disjoint Decoders.* In contrast to the previous configuration, distinct history embeddings, $\boldsymbol{h}_T$ for time and $\boldsymbol{h}_M$ for marks, are used to define $f^*(\tau;\boldsymbol{\theta}_T)$, $\lambda^*(t;\boldsymbol{\theta}_T)$, $\Lambda^*(t;\boldsymbol{\theta}_T)$, and $p^*(k|\tau;\boldsymbol{\theta}_M)$ separately. This separation allows for the independent training of the time prediction and mark prediction tasks as described in Section 4.3. Models trained within this setting are labeled with a "++" symbol, e.g., THP++.

Compared to the base models, these configurations allow us to assess the impacts of (1) isolating the parameters for the decoders in the time and mark prediction tasks, and (2) using distinct history embeddings for each task, enabling fully disjoint training. A graphical illustration of these configurations is shown in Figure 3. We will often refer to these setups as base, base+, and base++ throughout the text. The distinction between LNM (RMTPP) and LNM+ (RMTPP+) stems from the modeling of the PMF of marks using our model in (9), which relaxes the conditional independence assumption inherent in the base model. To maintain a fair comparison, we ensure that each configuration controls for the number of parameters, keeping them roughly equivalent across settings to confirm that any observed performance improvements are not merely due to increased model capacity. Finally, all models are trained to minimize the average NLL given in (17)[6]. We provide further training details in Appendix E.

**Metrics.** To evaluate the performance of the different baselines on the time prediction task, we report the $\mathcal{L}_T$ term in (17) computed over all test sequences. Following (Dheur & Ben Taieb, 2023), we also quantify the (unconditional) probabilistic calibration of the fitted models by computing the Probabilistic Calibration Error (PCE). Finally, we evaluate the MAE in event inter-arrival time prediction. To this end, we predict the next $\tilde{\tau}$ as the median of the predicted distribution of inter-arrival times, i.e. $\tilde{\tau} = (F^*)^{-1}(0.5;\boldsymbol{\theta}_T)$, where the quantile function $(F^*)^{-1}(\,\cdot\,;\boldsymbol{\theta}_T)$ is estimated using a binary search algorithm.

Similarly, for the mark prediction task, we report the average $\mathcal{L}_M$ term in (17), and quantify the probabilistic calibration of the mark predictive distribution by computing the Expected Calibration Error (ECE) (Naeini et al., 2015), and through reliability diagrams (Guo et al., 2017; Kuleshov et al., 2018). Additionally, by predicting the mark of the next event as $\tilde{k} = \underset{k\in\mathbb{K}}{\arg\max}\, p^*(k|\tau;\boldsymbol{\theta}_M)$, we can assess the quality of the point predictions by means of various classification metrics. Specifically, we compute the Accuracy@$n$ for values of $n$ in $\{1,3,5\}$ and the Mean Reciprocal Rank (MRR) (Craswell, 2009) of mark predictions. Lower $\mathcal{L}_T$, $\mathcal{L}_M$, PCE and ECE is better, while higher Accuracy@$n$ and MRR is better.

---

[5]For the remainder of this paper, these models will be referred to as "base models".

[6]Note that for the base and base+ methods, the two terms in (17) are functions on shared parameters.

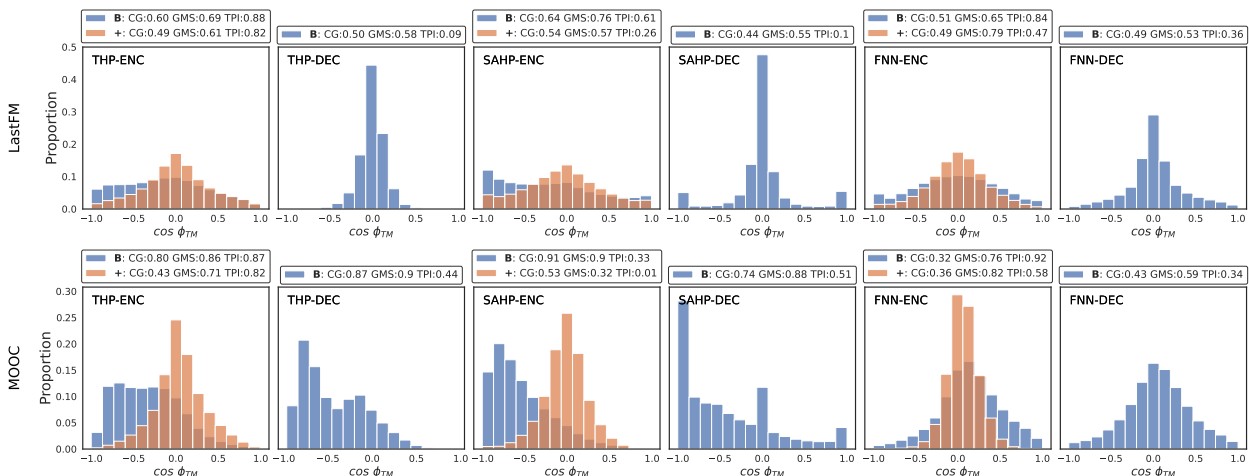

Figure 4: Distribution of cos $\phi_{TM}$ during training at the encoder (ENC) and decoder (DEC) heads for THP, SAHP and FNN in the base and base+ setup on LastFM and MOOC. "B" and "+" refer to the base and base+ models, respectively, and the distribution is obtained by pooling the values of $\phi_{TM}$ over 5 training runs. As the decoders are disjoint in the base+ setting, note that cos $\phi_{TM}$ is not defined.

## 6.1 Results and discussion

We present the $\mathcal{L}_T$ and $\mathcal{L}_M$ metrics for the base, base+, and base++ configurations across all datasets in Table 1. The PCE, MAE, ECE, MRR, and Accuracy@1,3,5 metrics, which reflect the subsequent discussion, are given in Appendix H.

**Distinct decoders mitigate gradient conflicts**. Based on the $\mathcal{L}_T$ and $\mathcal{L}_M$ metrics, we note a consistent improvement when moving from the base to the base+ setting for THP, SAHP, and FNN. This underscores the benefits of using two distinct decoders for time and mark prediction tasks with base+, leading to improved predictive accuracy compared to the base models. Figure 4 shows the distribution of cos $\phi_{TM}$ during training for THP, SAHP and FNN for both base and base+ on the LastFM and MOOC dataset, along with the average GMS and TPI for conflicting gradients. We would like too emphasize that both base and base+ share the *same* encoder architecture, which allows for a direct comparison of the distribution of cos $\phi_{TM}$ between the two settings during training. Appendix H provides detailed visualizations for other baselines and datasets. With the base model, a significant proportion of severe conflicts (as indicated by cos $\phi_{TM}$ in [-1, -0.5]) is often observed for the shared parameters of both encoder and decoder heads, typically with low GMS values. Additionally, with the base model, the TPI values suggest that these conflicting gradients at the encoder heads predominantly favor $\mathcal{T}_T$ (i.e., TPI > 0.5). In contrast, base+ inherently prevents conflicts at the decoder by separating the parameters for each task. During training with base+, there is also a noticeable reduction in the severity of conflicting gradients for the shared encoder parameters, as evidenced by a more concentrated distribution of cos $\phi_{TM}$ around 0. Moreover, the TPI values indicate that base+ generally achieves a more balanced training between both tasks, which further contributes to enhancing their individual performance. While we note that the GMS values do not consistently improve between the base and base+ settings, improvements with respect to $\mathcal{L}_T$ and $\mathcal{L}_M$ suggest that this effect is offset by a reduction in conflicts during training. Finally, while LNM and RMTPP already avoid decoder conflicts in their base models by decomposing the parameters, explicitly modeling the dependency of marks on time with base+ further enhances mark prediction performance.

**Disjoint training enhances mark prediction accuracy**. Returning to Table 1, moving from the base+ to the base++ setting often leads to improvements in the $\mathcal{L}_M$ metric for most baselines, while the $\mathcal{L}_T$ metric generally remains comparable between the two configurations. Although conflicting gradients are typically reduced when moving from base to base+, this pattern indicates that the residual conflicts primarily hinder the mark prediction task. In contrast, base++ effectively eliminates these conflicts by using distinct history representations for each task. A significant advantage of base++ is that it allows one task to continue training after the other has reached convergence. For example, Figure 5 illustrates the validation losses $\mathcal{L}_T$ and $\mathcal{L}_M$ for SAHP++ on MOOC. Thanks to disjoint training, the $\mathcal{L}_M$ metric

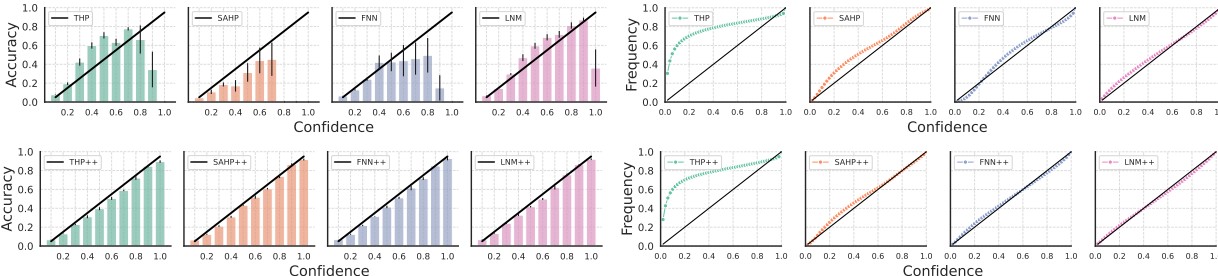

Figure 6: Reliability diagrams of $p^*(k|\tau;\boldsymbol{\theta})$ (left) and $f^*(\tau;\boldsymbol{\theta})$ (right) for the base models and their "++" extensions on the LastFM dataset. Frequency and accuracy aligning with the black diagonal correspond to perfect calibration. The reliability diagrams are averaged over 5 splits, and error bars refer to the standard error.

Table 1: $\mathcal{L}_T$ and $\mathcal{L}_M$ results of the different setups across all datasets. The values are computed over 5 splits, and the standard error is reported in parenthesis. Best results are highlighted in bold.

| | | | $\mathcal{L}_M$ | | | | | | $\mathcal{L}_T$ | | |
| | LastFM | MOOC | Github | Reddit | Stack O. | | LastFM | MOOC | Github | Reddit | Stack O. |
|---|---|---|---|---|---|---|---|---|---|---|---|
| THP | 714.0 (16.5) | 93.3 (1.4) | 128.3 (17.3) | **39.9 (1.3)** | 105.3 (0.7) | THP | -945.1 (41.3) | -135.9 (1.3) | -242.5 (38.5) | -72.3 (2.3) | -84.0 (1.4) |
| THP+ | 695.9 (15.7) | 76.9 (1.2) | 120.8 (16.0) | 42.7 (1.2) | 103.3 (0.7) | THP+ | -994.1 (48.8) | -130.6 (1.3) | -255.7 (42.0) | -85.5 (2.3) | **-84.7 (1.4)** |
| THP++ | **651.1 (18.1)** | **70.9 (1.0)** | **112.1 (15.4)** | 40.4 (1.3) | **103.0 (0.7)** | THP++ | **-1037.3 (47.0)** | **-136.0 (2.1)** | **-271.0 (50.8)** | **-87.9 (2.0)** | -84.2 (1.4) |
| STHP+ | 700.9 (16.6) | 83.2 (1.3) | 121.4 (15.8) | **43.7 (1.1)** | 104.0 (0.7) | STHP+ | -993.3 (44.2) | -128.2 (1.0) | -208.9 (32.2) | -76.6 (2.1) | **-83.6 (1.4)** |
| STHP++ | 696.5 (15.9) | 82.4 (1.4) | 114.7 (14.9) | 46.2 (0.8) | 105.6 (0.6) | STHP++ | **-1014.9 (42.1)** | **-132.9 (1.8)** | **-237.0 (39.3)** | **-78.2 (2.1)** | -83.3 (1.4) |
| SAHP | 825.5 (25.6) | 163.0 (2.2) | 138.0 (19.1) | 77.9 (4.7) | 108.4 (1.0) | SAHP | -1263.8 (57.9) | -266.0 (3.5) | -346.5 (57.4) | -72.9 (1.8) | -89.7 (1.4) |
| SAHP+ | 740.0 (24.6) | 73.8 (1.0) | 116.8 (15.2) | 43.4 (1.2) | 103.3 (0.7) | SAHP+ | **-1320.4 (58.4)** | -288.8 (3.5) | -358.1 (57.6) | -94.8 (2.3) | **-89.9 (1.4)** |
| SAHP++ | **654.8 (17.3)** | **71.0 (1.1)** | **114.1 (15.2)** | 40.5 (0.8) | 103.1 (0.7) | SAHP++ | -1320.0 (60.5) | **-293.8 (3.6)** | **-366.0 (59.3)** | **-95.4 (2.1)** | -77.4 (1.2) |
| LNM | 685.2 (15.8) | 86.6 (1.2) | 116.8 (15.2) | 43.4 (1.2) | 106.5 (0.7) | LNM | -1326.3 (55.8) | -310.0 (3.8) | -380.4 (59.8) | **-96.4 (2.0)** | -91.0 (1.4) |
| LNM+ | 668.6 (15.8) | 77.1 (1.3) | 112.3 (15.1) | 41.4 (1.1) | **103.2 (0.7)** | LNM+ | -1320.4 (60.5) | **-310.6 (3.7)** | -378.7 (59.4) | **-96.4 (2.0)** | **-91.1 (1.3)** |
| LNM++ | **637.2 (19.4)** | **73.8 (1.1)** | **111.5 (15.2)** | 40.8 (1.0) | 103.2 (0.7) | LNM++ | **-1334.7 (58.5)** | -307.6 (3.9) | **-381.2 (59.9)** | -96.3 (2.2) | -90.5 (1.4) |
| FNN | 739.5 (25.2) | 78.8 (1.3) | 113.5 (15.4) | 47.0 (1.4) | 107.3 (0.6) | FNN | -1276.2 (58.6) | -280.9 (3.2) | -363.1 (57.2) | -75.3 (2.5) | -81.2 (1.3) |
| FNN+ | 672.1 (17.9) | 72.3 (1.1) | 111.6 (15.1) | 41.2 (0.9) | 103.2 (0.7) | FNN+ | **-1324.6 (59.6)** | **-302.2 (3.6)** | -364.5 (57.0) | -94.4 (2.1) | **-90.0 (1.8)** |
| FNN++ | **648.6 (16.2)** | **71.8 (1.0)** | **109.5 (15.0)** | **40.1 (1.0)** | **103.1 (0.7)** | FNN++ | -1324.2 (58.1) | -300.9 (3.7) | **-365.0 (56.7)** | **-96.1 (2.2)** | -88.9 (1.4) |
| RMTPP | 684.6 (15.6) | 87.0 (1.2) | 126.4 (18.3) | 41.4 (0.9) | 106.5 (0.7) | RMTPP | -1052.9 (46.0) | -178.6 (1.8) | -268.0 (54.3) | **-88.0 (2.2)** | -83.3 (1.4) |
| RMTPP+ | 681.5 (16.3) | 74.9 (1.3) | 118.7 (15.4) | 42.0 (1.0) | **103.0 (0.7)** | RMTPP+ | -1040.7 (45.8) | **-187.8 (3.1)** | -272.2 (49.0) | -87.9 (2.0) | **-83.4 (1.4)** |
| RMTPP++ | **654.5 (16.7)** | **71.4 (1.1)** | **112.6 (15.3)** | 40.6 (1.2) | 103.1 (0.7) | RMTPP++ | **-1071.1 (50.3)** | -182.2 (1.9) | **-287.2 (52.6)** | -86.6 (2.0) | -82.9 (1.4) |

can be further optimized for additional epochs after training of the $\mathcal{L}_T$ metric ceases due to overfitting, thus achieving gains in mark prediction performance. This feature is absent in base and base+, where both $\mathcal{L}_M$ and $\mathcal{L}_T$ metrics rely on a shared set of parameters. In Figure 5, $\boldsymbol{\theta}_T$ is fixed after the vertical red line, resulting in a constant validation $\mathcal{L}_T$ for the remaining training epochs of the model.

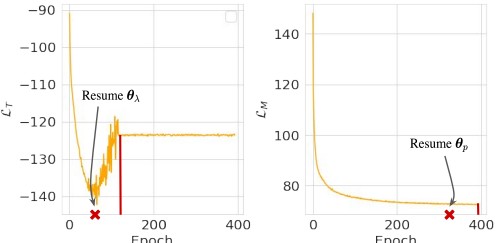

Finally, referring back to the preliminary experiment on Figure 2, the results with respect to $\mathcal{L}_T$ and $\mathcal{L}_M$ in Table 1 suggest that conflicting gradients are mostly detrimental to model performance if (1) they are in great proportion during training (i.e. high CG) and (2) if they are associated to low GMS values.

Figure 5: Validation curves of the $\mathcal{L}_T$ and $\mathcal{L}_M$ components for SAHP++ on MOOC.

For instance, on Figure 2, THP and SAHP exhibit high CG associated to high GMS values, whereas the remaining models conversely show a more balanced CG, but with lower GMS values. We find that model performance often improves in both these scenarios when preventing conflicting gradients altogether in the base++ setting.

**Reliability Diagrams**. Figure 6 presents the reliability diagrams for the predictive distributions of arrival-times and marks for base and base++ on LastFM. The diagrams show that the base++ models are generally better calibrated than their base counterparts, as evidenced by the bin accuracies aligning more closely with the diagonal. This improvement aligns with the results in Table 1, where a lower $\mathcal{L}_M$ indicates better accuracy. However, improvements in the calibration of arrival times between base and base++ models are less noticeable, suggesting that conflicting gradients during training predominantly affect the mark prediction task. We provide reliability diagrams for other baselines and datasets in Appendix H.

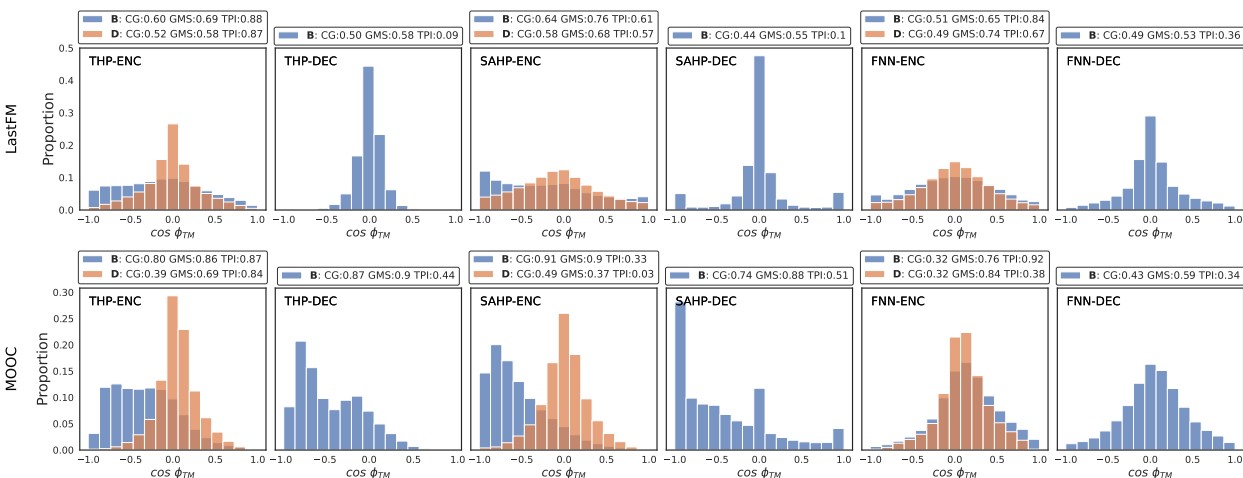

Figure 7: Distribution of $\cos\phi_{TM}$ during training at the encoder (ENC) and decoder (DEC) heads for THP, SAHP and FNN in the base and base-D setup on LastFM and MOOC. "B" and "D" refer to the base and base-D models, respectively, and the distribution is obtained by pooling the values of $\phi_{TM}$ over 5 training runs. As the decoders are disjoint in the base-D setting, note that $\cos\phi_{TM}$ is not defined.

Table 2: $\mathcal{L}_T$ and $\mathcal{L}_M$ results for the base, base-D, and base-DD settings across all datasets. The values are computed over 5 splits, and the standard error is reported in parenthesis. Best results are highlighted in bold.

| | | | $\mathcal{L}_M$ | | | | | | $\mathcal{L}_T$ | | |
| | LastFM | MOOC | Github | Reddit | Stack O. | | LastFM | MOOC | Github | Reddit | Stack O. |
|---|---|---|---|---|---|---|---|---|---|---|---|
| THP | 714.0 (16.5) | 93.3 (1.4) | 128.3 (17.3) | 39.9 (1.3) | 105.3 (0.7) | THP | -945.1 (41.3) | -135.9 (1.3) | -242.5 (38.5) | -72.3 (2.3) | -84.0 (1.4) |
| THP-D | 677.1 (19.4) | 82.8 (1.2) | 121.4 (16.8) | 39.1 (0.9) | 104.7 (0.7) | THP-D | -995.1 (50.2) | **-164.8 (1.8)** | -258.5 (47.2) | -90.3 (1.9) | **-85.3 (1.4)** |
| THP-DD | **607.2 (16.2)** | **79.8 (1.2)** | 113.5 (14.9) | **38.2 (1.0)** | 104.4 (0.6) | THP-DD | **-1023.8 (53.0)** | -162.1 (4.3) | **-279.0 (55.2)** | **-92.6 (2.1)** | -84.8 (1.4) |
| SAHP | 825.5 (25.6) | 163.0 (2.2) | 138.0 (19.1) | 77.9 (4.7) | 108.4 (1.0) | SAHP | -1263.8 (57.9) | -266.0 (3.5) | -346.5 (57.4) | -72.9 (1.8) | -89.7 (1.4) |
| SAHP-D | 832.1 (32.0) | 93.3 (1.5) | 128.8 (18.5) | 56.0 (1.0) | 105.1 (0.7) | SAHP-D | -1319.1 (57.9) | -288.7 (3.7) | -348.7 (58.2) | **-92.9 (2.3)** | **-90.1 (1.3)** |
| SAHP-DD | **692.1 (19.7)** | **89.0 (1.6)** | 115.4 (15.3) | 51.0 (0.5) | 104.8 (0.6) | SAHP-DD | **-1319.9 (59.3)** | **-294.0 (3.6)** | **-367.9 (59.1)** | -87.0 (3.5) | -87.1 (2.6) |
| FNN | 739.5 (25.2) | 78.8 (1.3) | 113.5 (15.4) | **47.0 (1.4)** | 107.3 (0.6) | FNN | -1276.2 (58.6) | -280.9 (3.2) | **-363.1 (57.2)** | -75.3 (2.5) | -81.2 (1.3) |
| FNN-D | 732.8 (19.7) | **76.8 (1.2)** | 112.9 (15.4) | 56.9 (0.9) | 103.8 (0.7) | FNN-D | **-1314.1 (59.7)** | -286.8 (3.5) | -360.0 (55.9) | -81.3 (1.7) | **-81.7 (1.3)** |
| FNN-DD | **670.0 (18.1)** | 79.7 (1.4) | **111.4 (15.3)** | 48.8 (1.6) | **103.7 (0.7)** | FNN-DD | -1294.4 (57.5) | -286.3 (3.8) | -362.5 (56.4) | **-82.7 (2.2)** | -81.4 (1.4) |

**Isolating the impact of conflicts on performance.** Our experiments reveal that the base+ and base++ settings result in a decrease of conflicting gradients and to enhanced performance with respect to the time and mark prediction tasks. However, for THP, SAHP and FNN, these settings do not enable us to disentangle performance gains brought by a decrease in conflicts from the ones brought by modifications of the decoder architecture. To address this limitation, we introduce the following two settings based on the duplicated model approach of Section 4.1:

(1) *Shared History Encoders and Duplicated Decoders.* The functions $\lambda^*(t;\boldsymbol{\theta}_T)/\Lambda^*(t;\boldsymbol{\theta}_T)$ and $p^*(k|t;\boldsymbol{\theta}_M)$ are obtained through (7) from two identical parametrizations of the same base decoder. However, similar to the base+ setting, a common history encoder $\boldsymbol{h}$ is used, meaning that these functions still share parameters via $\boldsymbol{\theta}_h$. Models trained in this setting are indicated with a "-D" sign, e.g. THP-D.

(2) *Disjoint History Encoders and Duplicated Decoders.* This setting differs from the previous one in the use of two distinct history embeddings $\boldsymbol{h}_T$ and $\boldsymbol{h}_M$ in (7), implying that $\lambda^*(t;\boldsymbol{\theta}_T)/\Lambda^*(t;\boldsymbol{\theta}_T)$ and $p^*(k|t;\boldsymbol{\theta}_M)$ are now completely disjoint parametrizations. We use the label "-DD" to denote the models trained in this setting, e.g. THP-DD.

In contrast to base+ and base++, the base-D and base-DD settings retain the same architecture as the base model, enabling us to directly evaluate the impact of conflicting gradients on performance. In Table 2, we report the performance with respect to $\mathcal{L}_T$ and $\mathcal{L}_M$ for THP, SAHP, and FNN trained in the base, base-D and base-DD settings. We follow the same experimental setup as before, and maintain the number of parameters comparable between the different settings for fair comparison. We almost systematically observe improvements on both tasks when moving from the base to the base-D or base-DD settings. Moreover, these performance gains are often associated to a decrease in (severe) conflicts during training, as shown on Figure 7. Furthermore, when comparing the results between Tables 1 and 2, we note that our base+ and base++ parametrizations often show improved performance compared to the base-D and base-DD settings, especially

on the mark prediction task. This highlights that the benefits of our parametrizations extend beyond the prevention of conflicts to achieve greater predictive performance. We provide more visualizations in Appendix H.6.

## 7 Conclusion, Limitations, and Future Work

Learning a neural MTPP model can be essentially interpreted as a two-task learning problem, in which one task is focused on learning a predictive distribution of arrival times, and the other on learning a predictive distribution of event types, known as marks. Typically, most neural MTPP models implicitly require these two tasks to share a common set of trainable parameters. In this paper, we demonstrate that this parameter sharing leads to the emergence of conflicting gradients during training, often resulting in degraded performance on each individual task. To prevent this issue, we introduce novel parametrizations of neural MTPP models that enable separate modeling and training of each task, effectively preventing the occurrence of conflicting gradients. Through extensive experiments on real-world event sequence datasets, we validate the advantages of our framework over the original model configurations, particularly in the context of mark prediction. However, we acknowledge several limitations in our study. Firstly, our focus was solely on categorical marks. Investigating conflicting gradients in more complex scenarios, such as temporal graphs (Trivedi et al., 2019; Gracious & Dukkipati, 2023) or spatio-temporal point processes (Zhou & Yu, 2023; Zhang et al., 2023), presents a promising avenue for future research. Secondly, our analysis was limited to neural MTPP models trained using the negative log-likelihood. Extending our framework to other proper scoring rules (Brehmer et al., 2023) is also a potential area for future exploration.

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

# A    Additional Forms of the Negative Log-Likelihood

Consider a dataset $\mathcal{S} = \{\mathcal{S}_1, ..., \mathcal{S}_L\}$, where each sequence $\mathcal{S}_l$ comprises $n_l$ events with arrival times observed within the interval $[0, T]$ and $l = 1, ..., L$. The average sequence negative log-likelihood (NLL) for these sequences can be expressed as a function of the marked intensity $\lambda^*(t; \boldsymbol{\theta}_T)$ or the compensator $\Lambda^*(t; \boldsymbol{\theta}_T)$ as follows:

$$\mathcal{L}(\boldsymbol{\theta}_T, \boldsymbol{\theta}_M; \mathcal{S}) = \underbrace{-\frac{1}{L}\sum_{l=1}^{L}\sum_{i=1}^{n_l}\log\,\lambda^*(t_{l,i}; \boldsymbol{\theta}_T) + \int_{t_n}^{T}\lambda^*(s; \boldsymbol{\theta}_T)ds}_{\mathcal{L}_T(\boldsymbol{\theta}_T; \mathcal{S})} \underbrace{-\frac{1}{L}\sum_{l=1}^{L}\sum_{i=1}^{n_l}\log\,p^*(k_{l,i}|\tau_{l,i}; \boldsymbol{\theta}_M)}_{\mathcal{L}_M(\boldsymbol{\theta}_M; \mathcal{S})}, \quad (18)$$

and

$$\mathcal{L}(\boldsymbol{\theta}_T, \boldsymbol{\theta}_M; \mathcal{S}) = \underbrace{-\frac{1}{L}\sum_{l=1}^{L}\left[\sum_{i=1}^{n_l}\log\,\left[\frac{d}{dt}\Lambda^*(t_{l,i}; \boldsymbol{\theta}_T)\right]\right] + \Lambda(T; \boldsymbol{\theta}_T))}_{\mathcal{L}_T(\boldsymbol{\theta}_T; \mathcal{S})} \underbrace{-\frac{1}{L}\sum_{l=1}^{L}\sum_{i=1}^{n_l}\log\,p^*(k_{l,i}|\tau_{l,i}; \boldsymbol{\theta}_M)}_{\mathcal{L}_M(\boldsymbol{\theta}_M; \mathcal{S})}. \quad (19)$$

# B    Alternative Scoring Rules

The NLL in (1) has been largely adopted as the default scoring rule for learning MTPP models (Shchur et al., 2021). However, our framework can be extended even further by using alternative scoring rules other than the NLL for assessing the mark and time prediction tasks. Let $S^f$, $S^m$ and $S^w$ be (strictly) consistent scoring rules for $f^*(\tau; \boldsymbol{\theta}_T)$, $p^*(k|\tau; \boldsymbol{\theta}_M)$ and $1 - F^*(T; \boldsymbol{\theta}_T)$, respectively. Given a sequence $\mathcal{S}$ of $n$ events (i.e. $L = 1$), the scoring rule

$$\mathcal{L}(\boldsymbol{\theta}; \mathcal{S}) = \underbrace{\sum_{i=1}^{n}\left[S^f(f^*(\tau_i; \boldsymbol{\theta}_T))\right] + S^w\left(1 - F^*(T; \boldsymbol{\theta}_T)\right)}_{\mathcal{L}_T(\boldsymbol{\theta}_T, \mathcal{S})} + \underbrace{\sum_{i=1}^{n}S^p(p^*(k_i|\tau_i; \boldsymbol{\theta}_M))}_{\mathcal{L}_M(\boldsymbol{\theta}_M, \mathcal{S})}, \quad (20)$$

is (strictly) consistent for the conditional joint density $f^*(\tau, k; \boldsymbol{\theta}_T, \boldsymbol{\theta}_M)$ restricted to the interval $[0, T]$ (Brehmer et al., 2023). Using the LogScore for $S^f$, $S^m$ and $S^w$ in equation 20 reduces to the NLL in (17). One can also use other choices tailored to the specific task. For instance, one can choose to use the continuous ranked probability score (CRPS) (Gneiting & Raftery, 2007) for $S^f$ to evaluate the predictive distribution of inter-arrival times (Ben Taieb, 2022). Similarly, the Brier score (Brier, 1950) can be used for $S^p$ to evaluate the predictive distribution of marks. Contrary to the *local* property of the LogScore, both the CRPS and the Brier score are *sensitive to distance*, in the sense that they reward predictive distributions that assign probability mass close to the observed realization (Gebetsbergera et al., 2018). Nonetheless, the choice between local and non-local proper scoring rules has been generally subjective in the literature. While exploring alternative scoring rules to train neural MTPP models is an exciting research direction, we leave it as future work and train the models exclusively on the NLL in (17).

# C    Proof of Corollary 1

Consider a base model $\lambda_k^*(t; \boldsymbol{\theta})$ with trainable parameters $\boldsymbol{\theta}$, and a disjoint parametrization of $\lambda^*(t; \boldsymbol{\theta}_T)$ and $p^*(k|t; \boldsymbol{\theta}_M)$ obtained in (7) from two identical $\lambda_k^*(t; \boldsymbol{\theta}_T)$ and $\lambda_k^*(t; \boldsymbol{\theta}_M)$ with trainable parameters $\boldsymbol{\theta}_T$ and $\boldsymbol{\theta}_M$, respectively. The NLL losses for $\boldsymbol{\theta}$ and $\{\boldsymbol{\theta}_T, \boldsymbol{\theta}_M\}$ respectively write

$$\mathcal{L}(\boldsymbol{\theta}) = \mathcal{L}_T(\boldsymbol{\theta}) + \mathcal{L}_M(\boldsymbol{\theta}), \quad (21)$$

$$\mathcal{L}(\{\boldsymbol{\theta}_T, \boldsymbol{\theta}_M\}) = \mathcal{L}_T(\boldsymbol{\theta}_T) + \mathcal{L}_M(\boldsymbol{\theta}_M). \quad (22)$$

Suppose that at training iteration $s \in \mathbb{N}$, $\boldsymbol{\theta}$, $\boldsymbol{\theta}_T$ and $\boldsymbol{\theta}_M$ are all initialized with the same $\boldsymbol{\theta}^s$. This implies that $\lambda_k^*(t; \boldsymbol{\theta})$, $\lambda_k^*(t; \boldsymbol{\theta}_T)$ and $\lambda_k^*(t; \boldsymbol{\theta}_M)$ are all identical. Assuming that $\mathcal{L}_T$ and $\mathcal{L}_M$ are differentiable, the gradient update steps for $\boldsymbol{\theta}$, $\boldsymbol{\theta}_T$ and $\boldsymbol{\theta}_M$ are

$$\boldsymbol{\theta}^{s+1} = \boldsymbol{\theta}^s - \alpha(\boldsymbol{g}_T + \boldsymbol{g}_M), \quad \boldsymbol{\theta}_T^{s+1} = \boldsymbol{\theta}^s - \alpha\boldsymbol{g}_T \quad \text{and} \quad \boldsymbol{\theta}_M^{s+1} = \boldsymbol{\theta}^s - \alpha\boldsymbol{g}_M, \quad (23)$$

where $\alpha > 0$ is the learning rate and

$$\boldsymbol{g}_T = \nabla_{\boldsymbol{\theta}} \mathcal{L}_T\left(\boldsymbol{\theta}^s\right) = \nabla_{\boldsymbol{\theta}_T} \mathcal{L}_T\left(\boldsymbol{\theta}^s\right) \quad \text{and} \quad \boldsymbol{g}_M = \nabla_{\boldsymbol{\theta}} \mathcal{L}_M\left(\boldsymbol{\theta}^s\right) = \nabla_{\boldsymbol{\theta}_M} \mathcal{L}_M\left(\boldsymbol{\theta}^s\right). \tag{24}$$

Denoting $\phi_{TM}$ as the angle between $\boldsymbol{g}_T$ and $\boldsymbol{g}_M$, we have the following corollary of Theorem 4.1. from (Shi et al., 2023):

**Corollary 1.** *Assume that $\mathcal{L}_T$ and $\mathcal{L}_M$ are differentiable, and that the learning rate $\alpha$ is sufficiently small. If $\cos \phi_{TM} < 0$, then $\mathcal{L}(\{\boldsymbol{\theta}_T^{s+1}, \boldsymbol{\theta}_M^{s+1}\}) < \mathcal{L}(\boldsymbol{\theta}^{s+1})$.*

*Proof.* Let us consider the first order Taylor approximations of the total loss $\mathcal{L} = \mathcal{L}_T + \mathcal{L}_M$ near $\boldsymbol{\theta}^{s+1}$ and $\{\boldsymbol{\theta}_T^{s+1}, \boldsymbol{\theta}_M^{s+1}\}$, respectively:

$$\mathcal{L}(\boldsymbol{\theta}^{s+1}) = \mathcal{L}(\boldsymbol{\theta}^s) + (\boldsymbol{\theta}^{s+1} - \boldsymbol{\theta}^s)^{\mathsf{T}} \boldsymbol{g}_T + (\boldsymbol{\theta}^{s+1} - \boldsymbol{\theta}^s)^{\mathsf{T}} \boldsymbol{g}_M + o(\alpha). \tag{25}$$

$$\mathcal{L}(\{\boldsymbol{\theta}_T^{s+1}, \boldsymbol{\theta}_M^{s+1}\}) = \mathcal{L}(\boldsymbol{\theta}^s) + (\boldsymbol{\theta}_T^{s+1} - \boldsymbol{\theta}^s)^{\mathsf{T}} \boldsymbol{g}_T + (\boldsymbol{\theta}_M^{s+1} - \boldsymbol{\theta}^s)^{\mathsf{T}} \boldsymbol{g}_M + o(\alpha). \tag{26}$$

Taking the difference between $\mathcal{L}(\{\boldsymbol{\theta}_T^{s+1}, \boldsymbol{\theta}_M^{s+1}\})$ and $\mathcal{L}(\boldsymbol{\theta}^{s+1})$ yields

$$\mathcal{L}(\{\boldsymbol{\theta}_T^{s+1}, \boldsymbol{\theta}_M^{s+1}\}) - \mathcal{L}(\boldsymbol{\theta}^{s+1}) = (\boldsymbol{\theta}_T^{s+1} - \boldsymbol{\theta}^{s+1})^{\mathsf{T}} \boldsymbol{g}_T + (\boldsymbol{\theta}_M^{s+1} - \boldsymbol{\theta}^{s+1})^{\mathsf{T}} \boldsymbol{g}_M + o(\alpha) \tag{27}$$

$$= \alpha \boldsymbol{g}_M^{\mathsf{T}} \boldsymbol{g}_T + \alpha \boldsymbol{g}_T^{\mathsf{T}} \boldsymbol{g}_M + o(\alpha) \tag{28}$$

$$= 2\alpha \|\boldsymbol{g}_T\| \|\boldsymbol{g}_M\| \cos \phi_{TM} + o(\alpha). \tag{29}$$

Provided that $\alpha$ is sufficiently small, this difference is negative if $\cos \phi_{TM}^d < 0$, where $\phi_{TM}$ is the angle between $\boldsymbol{g}_T$ and $\boldsymbol{g}_M$. $\qquad\square$

## D  Datasets

We use 5 real-world event sequence datasets for our experiments:

- **LastFM** (Hidasi & Tikk, 2012): Each sequence corresponds to a user listening to music records over time. The artist of the song is the mark.

- **MOOC** (Kumar et al., 2019): Records of students' activities on an online course system. Each sequence corresponds to a student, and the mark is the type of activity performed.

- **Github** (Trivedi et al., 2019): Actions of software developers on the open-source platform Github. Each sequence corresponds a developer, and the mark is the action performed (e.g. fork, pull request,...).

- **Reddit** (Kumar et al., 2019): Sequences of posts to sub-reddits that users make on the social website Reddit. Each sequence is a user, and the sub-reddits to which the user posts is considered as the mark.

- **Stack Overflow** (Du et al., 2016). Sequences of badges that users receive over time on the website Stack Overflow. A sequence is a specific user, and the type of badge received is the mark.

We employ the pre-processed version of these datasets as described in (Bosser & Ben Taieb, 2023) which can be openly accessed at this url: `https://www.dropbox.com/sh/maq7nju7v5020kp/AAAFBvzxeNqySRsAm-zgU7s3a/processed/data?dl=0&subfolder_nav_tracking=1` (MIT License). Specifically, each dataset is filtered to contain the 50 most represented marks, and all arrival-times are rescaled in the interval [0,10] to avoid numerical instabilities. To save computational time, the number of sequences in Reddit is reduced by 50%. Each dataset is randomly partitioned into 3 train/validation/test splits (60%/20%/20%). The summary statistics for each (filtered) dataset is reported in Table 3.

Table 3: Datasets statistics

|  | #Seq. | #Events | Mean Length | Max Length | Min Length | #Marks |
|---|---|---|---|---|---|---|
| LastFM | 856 | 193441 | 226.0 | 6396 | 2 | 50 |
| MOOC | 7047 | 351160 | 49.8 | 416 | 2 | 50 |
| Github | 173 | 20656 | 119.4 | 4698 | 3 | 8 |
| Reddit | 4278 | 238734 | 55.8 | 941 | 2 | 50 |
| Stack Overflow | 7959 | 569688 | 71.6 | 735 | 40 | 22 |

# E   Training details

**Training details.**   For all models, we minimize the average NLL in (17) on the training sequences using mini-batch gradient descent with the Adam optimizer (Kingma & Ba, 2014) and a learning rate of $10^{-3}$. For the base models and the base+ setup, an early-stopping protocol interrupts training if the model fails to show improvement in the total validation loss (i.e., $\mathcal{L}_T + \mathcal{L}_T$) for 50 consecutive epochs. Conversely, in the base++ setup, two distinct early-stopping protocols are implemented for the $\mathcal{L}_T$ and $\mathcal{L}_M$ terms, respectively. If one of these terms does not show improvement for 50 consecutive epochs, we freeze the parameters of the associated functions (e.g. $\boldsymbol{\theta}_T$ for $f^*(\tau; \boldsymbol{\theta}_T)$) and allow the remaining term to continue training. Training is ultimately interrupted if both early-stopping criteria are met.

In all setups, the optimization process can last for a maximum of 500 epochs, and we revert the model parameters to their state with the lowest validation loss after training. Finally, we evaluate the model by computing test metrics on the test sequences of each split. Our framework is implemented in a unified codebase using PyTorch[7]. All models were trained on a machine equipped with an AMD Ryzen Threadripper PRO 3975WX CPU running at 4.1 GHz and a Nvidia RTX A4000 GPU.

**Encoding past events.**   To obtain the encoding $\boldsymbol{e}_i \in \mathbb{R}^{d_e}$ of an event $e_i = (t_i, k_i)$ in $\mathcal{H}_t$, we follow the work of (Enguehard et al., 2020) by first mapping $t_i$ to a vector of sinusoidal functions:

$$\boldsymbol{e}_i^t = \bigoplus_{j=0}^{d_t/2-1} \sin\,(\alpha_j t_i) \oplus \cos\,(\alpha_j t_i) \in \mathbb{R}^{d_t}, \tag{30}$$

where $\alpha_j \propto 1000^{\frac{-2j}{d_t}}$ and $\oplus$ is the concatenation operator. Then, a mark embedding $\boldsymbol{e}_i^k \in \mathbb{R}^{d_k}$ for $k_i$ is generated as $\boldsymbol{e}_i^k = \mathbf{E}^k \boldsymbol{k}_i$, where $\mathbf{E}^k \in \mathbb{R}^{d_k \times K}$ is a learnable embedding matrix, and $\boldsymbol{k}_i \in \{0,1\}^K$ is the one-hot encoding of $k_i$. Finally, we obtain $\boldsymbol{e}_i$ through concatenation, i.e. $\boldsymbol{e}_i = [\boldsymbol{e}_i^t || \boldsymbol{e}_i^k]$.

**Hyperparameters.**   To ensure that changes in performance are solely attributed to the features enabled by our framework, we control the number of parameters such the a baseline's capacity remains equivalent across the base, base+, base++, base-D, and base-DD setups. Notably, since STHP inherently models the decomposition of the marked intensity, the base and base+ configurations are equivalent. Also, LNM, RMTPP and STHP are equivalent between the base+ and the base-D settings, and between the base++ and base-DD settings. Hence, we only consider these models in the base+ and base++ settings. Furthermore, Table 4 provides the total number of trainable parameters for each setup when trained on the LastFM dataset, as well as their distribution across the encoder and decoder heads. For all baselines and setups, we use a single encoder layer, and the dimension $d_e$ of the event encodings is set to 8. Additionally, we chose a value of $M = 32$ for the number of mixture components. It is worth noting that (Shchur et al., 2020) found LogNormMix to be robust to the choice of $M$. Finally, we set the number of GCIF projections to $C = 32$.

# F   Computational Time

We report in Table equation 5 the average execution time (in seconds) for a single forward and backward pass on all training sequences of all datasets. The results are averaged over 50 epochs. We notice that the computation of two separate embeddings $\boldsymbol{h}_T$ and $\boldsymbol{h}_M$ in the base++ setup inevitably leads to an increase

---

[7]https://pytorch.org/

Table 4: Number of parameters for each baseline when trained on the LastFM dataset. The distribution of parameters between the encoder and decoder heads is reported in parenthesis.

| | THP | STHP | SAHP | FNN | LNM | RMTPP |
|---|---|---|---|---|---|---|
| Base | 14720 (0.66/0.34) | \ | 15588 (0.68/0.32) | 15939 (0.65/0.35) | 13930 (0.67/0.33) | 13619 (0.69/0.31) |
| Base+ | 14786 (0.64/0.36) | 18586 (0.5/0.5) | 15210 (0.67/0.33) | 16083 (0.65/0.35) | 13946 (0.67/0.33) | 13669 (0.69/0.31) |
| Base++ | 14602 (0.63/0.37) | 18402 (0.5/0.5) | 15514 (0.67/0.33) | 14961 (0.67/0.33) | 13340 (0.66/0.34) | 13063 (0.67/0.33) |
| Base-D | 15512 (0.66/0.34) | \ | 15412 (0.67/0.33) | 16083 (0.65/0.35) | \ | \ |
| Base-DD | 15252 (0.66/0.34) | \ | 15464 (0.68/0.32) | 10446 (0.65/0.35) | \ | \ |

in execution time, which appears more pronounced for larger datasets such as Reddit and Stack Overflow. However, the increased computational complexity is generally offset by improved model performance, as detailed in Section 6.1.

Table 5: Average execution time (in seconds) for a single forward and backward pass on all training sequences for all datasets. Results are averaged over 50 epochs.

| | LNM | | | RMTPP | | | FNN | | | THP | | | SAHP | | | STHP | | |
|---|---|---|---|---|---|---|---|---|---|---|---|---|---|---|---|---|---|---|
| | Base | + | ++ | Base | + | ++ | Base | + | ++ | Base | + | ++ | Base | + | ++ | Base | + | ++ |
| LastFM | 2.2 | 2.23 | 3.26 | 1.91 | 1.94 | 2.94 | 4.29 | 3.66 | 4.74 | 2.96 | 2.57 | 3.62 | 4.82 | 3.42 | 4.48 | 3.08 | \ | 4.17 |
| MOOC | 3.97 | 3.99 | 5.6 | 3.24 | 3.33 | 4.37 | 6.36 | 6.28 | 8.07 | 5.19 | 4.15 | 5.76 | 8.08 | 5.91 | 7.74 | 5.61 | \ | 6.72 |
| Github | 0.34 | 0.39 | 0.51 | 0.34 | 0.34 | 0.5 | 0.76 | 0.65 | 0.8 | 0.38 | 0.43 | 0.58 | 0.51 | 0.55 | 0.69 | 0.49 | \ | 0.64 |
| Reddit | 8.74 | 9.53 | 11.5 | 7.99 | 8.14 | 11.79 | 19.13 | 16.47 | 20.32 | 9.75 | 9.78 | 13.5 | 13.63 | 11.99 | 15.9 | 11.59 | \ | 15.54 |
| Stack O. | 16.02 | 16.63 | 22.98 | 14.08 | 14.54 | 20.31 | 32.83 | 28.15 | 33.95 | 16.32 | 16.81 | 23.1 | 21.11 | 20.77 | 26.45 | 19.84 | \ | 26.07 |

# G    An Alternative Approach to Model the Joint Distribution

As detailed in Section (4.2), our parametrization of LNM+ alleviates the conditional independence of arrival-times and marks made in LNM (Shchur et al., 2020). Relatedly, Waghmare et al. (2022) also proposed an extension of LNM that relaxes this assumption, although their approach differs from ours in some key aspects. Specifically, their work parametrizes $f^*(\tau|k; \boldsymbol{\theta}_T)$ as a distinct mixture of log-normal distributions for each mark $k$, and $p^*(k; \boldsymbol{\theta}_M)$ is obtained by removing the temporal dependency in (9)[8]. For further reference, we denote this model as LNM-Joint. Although both LNM+ and LNM-Joint aim to model the joint distribution $f^*(\tau, k)$, some conceptual differences separate the two approaches:

1. By design, LNM-Joint cannot be trained in the base++ setup as it prevents the decomposition of the NLL into disjoint $\mathcal{L}_T$ and $\mathcal{L}_M$ terms. Indeed, suppose that we use two distinct history representations $\boldsymbol{h}_T$ and $\boldsymbol{h}_M$ to parametrize $f^*(\tau|k; \boldsymbol{\theta}_T)$ and $p^*(k; \boldsymbol{\theta}_M)$ respectively, as detailed in Section 4.2. Here, $\boldsymbol{\theta}_T$ and $\boldsymbol{\theta}_M$ are disjoint set of learnable parameters. The NLL of a training sequence $\mathcal{S} = \{(\tau_i, k_i)\}_{i=1}^n$ observed in $[0, T]$ would write

$$\mathcal{L}(\boldsymbol{\theta}_T, \boldsymbol{\theta}_M; \mathcal{S}) = \underbrace{-\sum_{i=1}^n \log f^*(\tau_i|k_i; \boldsymbol{\theta}_T) - \log\ (1 - F^*\ (T; \boldsymbol{\theta}_T, \boldsymbol{\theta}_M))}_{\mathcal{L}_T(\boldsymbol{\theta}_T, \boldsymbol{\theta}_M; \mathcal{S})} \underbrace{-\sum_{i=1}^n \log p^*(k_i; \boldsymbol{\theta}_M)}_{\mathcal{L}_M(\boldsymbol{\theta}_M; \mathcal{S})}, \quad (31)$$

where $F^*\ (T; \boldsymbol{\theta}_T; \boldsymbol{\theta}_M) = \int_0^{T-t_n} \sum_{k=1}^K f^*(s|k; \boldsymbol{\theta}_T)p^*(k; \boldsymbol{\theta}_M)ds$ depends on both $\boldsymbol{\theta}_T$ and $\boldsymbol{\theta}_M$. Consequently, the NLL cannot be disentangled into disjoint $\mathcal{L}_T$ and $\mathcal{L}_M$ terms, proscribing disjoint training in the base++ setup. Conversely, choosing to parametrize $f^*(\tau; \boldsymbol{\theta}_T)$ and $p^*(k|\tau; \boldsymbol{\theta}_M)$ as done in our framework leads to the decomposition in (17) as $F^*(T; \boldsymbol{\theta}_T)$ is solely function of $\boldsymbol{\theta}_T$.

---

[8]Both $f^*(\tau|k; \boldsymbol{\theta}_T)$ and $p^*(k; \boldsymbol{\theta}_M)$ rely on a common history embedding $\boldsymbol{h}$.

2. For LNM-Joint, $M$ mixtures must be defined for each $k$, leading to $M \times K$ log-normal distributions in total. Conversely, in LNM+, $f^*(\tau; \boldsymbol{\theta}_T)$ does not scale with $K$, and $p^*(\tau|k; \boldsymbol{\theta}_M)$ requires an equivalent number of parameters as $p^*(k; \boldsymbol{\theta}_M)$ in LNM-Joint.

For completeness, we integrate LNM-Joint in our code base using the original implementation as reference. In Table (6), we compare its performance against LNM+ on the time and mark prediction tasks in terms of the $\mathcal{L}_T$, $\mathcal{L}_M$, and accuracy metrics, following the experimental setup detailed previously. We observe improved performance of LNM+ compared to LNM-Joint on the mark prediction task ($\mathcal{L}_M$ and accuracy), and competitive results on the time prediction task ($\mathcal{L}_T$). Despite both approaches modelling the joint distribution, our results suggest that the dependency between arrival times and marks is more accurately captured by $p^*(k|\tau; \boldsymbol{\theta}_M)$ than by $f^*(\tau|k; \boldsymbol{\theta}_T)$.

Table 6: $\mathcal{L}_T$, $\mathcal{L}_M$ and Accuracy results for LNM+ and LNM-Joint on all datasets. The values are computed over 3 splits, and the standard error is reported in parenthesis. Best results are highlighted in bold.

| | LastFM | MOOC | Github | Reddit | Stack O. |
|---|---|---|---|---|---|
| | | | $\mathcal{L}_M$ | | |
| LNM+ | **668.6 (15.8)** | **77.1 (1.3)** | **112.3 (15.1)** | **41.4 (1.1)** | **103.2 (0.7)** |
| Joint-LNM | 671.3 (17.1) | 127.0 (6.4) | 117.3 (15.5) | 42.9 (0.9) | 106.6 (0.7) |
| | | | Accuracy | | |
| LNM+ | **0.24 (0.01)** | **0.52 (0.0)** | **0.67 (0.01)** | **0.82 (0.0)** | **0.49 (0.0)** |
| Joint-LNM | 0.23 (0.01) | 0.23 (0.03) | 0.64 (0.01) | **0.82 (0.0)** | 0.47 (0.0) |
| | | | $\mathcal{L}_T$ | | |
| LNM+ | -1320.4 (60.5) | **-310.6 (3.7)** | -378.7 (59.4) | **-96.4 (2.0)** | **-91.1 (1.3)** |
| Joint-LNM | **-1326.2 (58.3)** | -303.9 (3.3) | **-381.0 (59.6)** | -94.3 (2.0) | -90.6 (1.4) |

# H    Additional results

## H.1    Evaluation metrics

Tables 7, 8 and 9 give the PCE, ECE, MRR, and accuracy@{1,3,5} for the base, base+ and base++ setups across all datasets. The metrics are averaged over 3 splits, and the standard error is given in parenthesis. We note that the results not discussed in the main text are consistent with our previous conclusions. Specifically, we observe general improvement with respect to mark related metrics (i.e. ECE, MRR, accuracy@{1,3,5}) when moving from the base models to the base+ or base++ setups. Finally, the PCE metric does not always improve between the base+ and ++ setups, suggesting that the remaining conflicting gradients at the encoder head in the base+ are mostly detrimental to the mark prediction task.

Finally, we report the results with respect to the MAE metric in Table 10. We notice that lower MAE values do not systematically match the lower values of $\mathcal{L}_T$ or PCE in Tables 1 and 7, indicating that the MAE may not be entirely appropriate to evaluate the time prediction task. As discussed in Shchur et al. (2021), neural MTPP models are probabilistic models that enable the generation of complete distributions over future events. In this context, point prediction metrics, like MAE, are deemed less suitable for evaluating MTPP models because they consider single point predictions into account. In contrast, the NLL and calibration scores directly evaluate the entire predictive distributions, and should be therefore favored compared to point prediction metrics.

## H.2    Reliability diagrams

Figures (8) to (10) show the reliability diagrams of the predictive distribution of arrival-times and marks for all models in the base and base++ setups on all datasets. In most cases, we observe improved mark calibration for the base++ setup compared to the base models, in accordance to the ECE results of Table 7. Additionally, improvements with respect to the calibration of the predictive distribution of arrival times is in general less prevalent, corroborating our discussion in the main text. This observation is also in coherent with the PCE results of Table 7. Nonetheless, we observe substantial time calibration improvements for SAHP and FNN when trained in the base++ setup on MOOC and Reddit.

Table 7: PCE and ECE results of the different setups across all datasets. The values are computed over 5 splits, and the standard error is reported in parenthesis. Best results are highlighted in bold.

| | PCE | | | | |
| | LastFM | MOOC | Github | Reddit | Stack O. |
|---|---|---|---|---|---|
| THP | 0.28 (0.0) | **0.37 (0.0)** | 0.2 (0.01) | 0.12 (0.0) | **0.01 (0.0)** |
| THP+ | **0.27 (0.01)** | **0.37 (0.0)** | **0.19 (0.01)** | 0.07 (0.0) | **0.01 (0.0)** |
| THP++ | 0.28 (0.01) | **0.37 (0.0)** | 0.19 (0.02) | **0.05 (0.0)** | **0.01 (0.0)** |
| STHP+ | **0.29 (0.01)** | 0.37 (0.0) | 0.29 (0.02) | **0.1 (0.0)** | **0.01 (0.0)** |
| STHP++ | **0.29 (0.01)** | 0.36 (0.0) | 0.25 (0.02) | **0.1 (0.0)** | **0.01 (0.0)** |
| SAHP | 0.06 (0.01) | 0.12 (0.0) | 0.07 (0.0) | 0.1 (0.01) | **0.01 (0.0)** |
| SAHP+ | 0.05 (0.01) | **0.03 (0.0)** | **0.04 (0.01)** | **0.01 (0.0)** | **0.0 (0.0)** |
| SAHP++ | **0.04 (0.01)** | **0.03 (0.0)** | **0.04 (0.01)** | 0.01 (0.0) | 0.04 (0.0) |
| LNM | **0.03 (0.01)** | **0.01 (0.0)** | **0.02 (0.0)** | **0.01 (0.0)** | **0.0 (0.0)** |
| LNM+ | 0.05 (0.01) | **0.01 (0.0)** | 0.03 (0.0) | **0.01 (0.0)** | **0.0 (0.0)** |
| LNM++ | **0.03 (0.0)** | **0.01 (0.0)** | 0.03 (0.0) | **0.01 (0.0)** | **0.0 (0.0)** |
| FNN | 0.04 (0.0) | 0.09 (0.0) | 0.04 (0.01) | 0.07 (0.0) | 0.05 (0.0) |
| FNN+ | 0.03 (0.0) | **0.01 (0.0)** | 0.03 (0.01) | **0.01 (0.0)** | **0.01 (0.0)** |
| FNN++ | **0.02 (0.0)** | **0.01 (0.0)** | **0.02 (0.0)** | **0.01 (0.0)** | **0.01 (0.0)** |
| RMTPP | **0.25 (0.01)** | 0.29 (0.0) | 0.18 (0.01) | **0.03 (0.0)** | **0.01 (0.0)** |
| RMTPP+ | 0.26 (0.01) | **0.27 (0.01)** | 0.18 (0.01) | **0.03 (0.0)** | **0.01 (0.0)** |
| RMTPP++ | 0.26 (0.0) | 0.29 (0.0) | **0.17 (0.01)** | 0.04 (0.0) | **0.01 (0.0)** |

| | ECE | | | | |
| | LastFM | MOOC | Github | Reddit | Stack O. |
|---|---|---|---|---|---|
| THP | 0.23 (0.03) | 0.07 (0.0) | 0.09 (0.02) | **0.02 (0.0)** | 0.04 (0.02) |
| THP+ | 0.05 (0.01) | **0.02 (0.0)** | **0.07 (0.01)** | 0.03 (0.01) | **0.01 (0.0)** |
| THP++ | **0.03 (0.0)** | **0.02 (0.0)** | 0.07 (0.02) | **0.02 (0.0)** | **0.01 (0.0)** |
| STHP+ | 0.05 (0.0) | **0.03 (0.0)** | 0.06 (0.02) | **0.03 (0.0)** | **0.02 (0.0)** |
| STHP++ | **0.04 (0.0)** | 0.04 (0.0) | 0.04 (0.01) | 0.05 (0.01) | 0.03 (0.01) |
| SAHP | 0.11 (0.01) | 0.13 (0.0) | 0.08 (0.02) | 0.08 (0.01) | 0.03 (0.0) |
| SAHP+ | 0.06 (0.01) | **0.02 (0.0)** | 0.07 (0.02) | 0.03 (0.01) | **0.01 (0.0)** |
| SAHP++ | **0.03 (0.0)** | **0.02 (0.01)** | 0.07 (0.01) | **0.02 (0.0)** | **0.01 (0.0)** |
| LNM | 0.09 (0.02) | 0.07 (0.01) | **0.05 (0.01)** | 0.03 (0.01) | 0.03 (0.01) |
| LNM+ | 0.05 (0.01) | 0.04 (0.01) | 0.06 (0.01) | **0.02 (0.0)** | **0.01 (0.0)** |
| LNM++ | **0.02 (0.0)** | **0.02 (0.0)** | **0.05 (0.01)** | **0.02 (0.0)** | **0.01 (0.0)** |
| FNN | 0.08 (0.01) | 0.05 (0.0) | **0.05 (0.0)** | 0.04 (0.01) | 0.02 (0.0) |
| FNN+ | 0.04 (0.01) | **0.02 (0.0)** | 0.06 (0.0) | **0.02 (0.0)** | **0.01 (0.0)** |
| FNN++ | **0.03 (0.0)** | **0.02 (0.0)** | 0.07 (0.02) | **0.02 (0.0)** | **0.01 (0.0)** |
| RMTPP | 0.05 (0.01) | 0.06 (0.01) | 0.07 (0.0) | 0.03 (0.0) | 0.03 (0.02) |
| RMTPP+ | 0.05 (0.01) | 0.03 (0.01) | **0.06 (0.0)** | 0.03 (0.01) | **0.01 (0.0)** |
| RMTPP++ | **0.04 (0.01)** | **0.02 (0.0)** | 0.06 (0.01) | **0.02 (0.0)** | **0.01 (0.0)** |

Table 8: Accuracy@1 and Accuracy@3 results of the different setups across all datasets. The values are computed over 5 splits, and the standard error is reported in parenthesis. Best results are highlighted in bold.

| | Accuracy | | | | |
| | LastFM | MOOC | Github | Reddit | Stack O. |
|---|---|---|---|---|---|
| THP | 0.18 (0.01) | 0.4 (0.0) | 0.59 (0.02) | **0.83 (0.0)** | 0.48 (0.0) |
| THP+ | 0.21 (0.01) | 0.52 (0.0) | 0.64 (0.01) | 0.82 (0.0) | **0.49 (0.0)** |
| THP++ | **0.25 (0.01)** | **0.55 (0.0)** | **0.67 (0.01)** | 0.82 (0.0) | **0.49 (0.0)** |
| STHP+ | 0.2 (0.01) | **0.46 (0.0)** | 0.63 (0.01) | **0.81 (0.0)** | 0.48 (0.0) |
| STHP++ | **0.21 (0.01)** | **0.46 (0.0)** | **0.65 (0.01)** | **0.81 (0.0)** | 0.48 (0.0) |
| SAHP | 0.05 (0.0) | 0.36 (0.01) | 0.6 (0.0) | 0.69 (0.02) | 0.48 (0.0) |
| SAHP+ | 0.14 (0.01) | 0.54 (0.0) | 0.65 (0.01) | **0.82 (0.0)** | **0.49 (0.0)** |
| SAHP++ | **0.24 (0.01)** | **0.56 (0.0)** | **0.67 (0.01)** | 0.82 (0.0) | **0.49 (0.0)** |
| LNM | 0.21 (0.01) | 0.43 (0.0) | 0.64 (0.01) | 0.81 (0.0) | 0.47 (0.0) |
| LNM+ | 0.24 (0.01) | 0.52 (0.0) | **0.67 (0.01)** | 0.82 (0.0) | **0.49 (0.0)** |
| LNM++ | **0.25 (0.01)** | 0.54 (0.0) | **0.67 (0.01)** | 0.82 (0.0) | 0.48 (0.0) |
| FNN | 0.13 (0.01) | 0.51 (0.0) | **0.67 (0.01)** | 0.8 (0.01) | 0.48 (0.0) |
| FNN+ | 0.23 (0.01) | **0.55 (0.0)** | **0.67 (0.01)** | 0.82 (0.0) | **0.49 (0.0)** |
| FNN++ | **0.25 (0.01)** | **0.55 (0.0)** | **0.67 (0.01)** | 0.82 (0.0) | **0.49 (0.0)** |
| RMTPP | 0.23 (0.01) | 0.42 (0.0) | 0.61 (0.02) | **0.82 (0.0)** | 0.47 (0.0) |
| RMTPP+ | 0.23 (0.01) | 0.53 (0.0) | 0.64 (0.01) | **0.82 (0.0)** | **0.49 (0.0)** |
| RMTPP++ | **0.24 (0.01)** | **0.55 (0.0)** | **0.67 (0.01)** | 0.82 (0.0) | **0.49 (0.0)** |

| | Accuracy@3 | | | | |
| | LastFM | MOOC | Github | Reddit | Stack O. |
|---|---|---|---|---|---|
| THP | 0.36 (0.01) | 0.75 (0.0) | 0.86 (0.01) | **0.91 (0.0)** | 0.82 (0.0) |
| THP+ | 0.37 (0.01) | 0.8 (0.0) | 0.86 (0.0) | 0.9 (0.0) | **0.83 (0.0)** |
| THP++ | **0.42 (0.01)** | **0.82 (0.0)** | 0.89 (0.0) | 0.9 (0.01) | **0.83 (0.0)** |
| STHP+ | 0.36 (0.01) | **0.78 (0.0)** | 0.88 (0.01) | **0.89 (0.0)** | **0.83 (0.0)** |
| STHP++ | 0.37 (0.01) | **0.78 (0.0)** | **0.89 (0.0)** | **0.89 (0.0)** | 0.82 (0.0) |
| SAHP | 0.14 (0.01) | 0.57 (0.0) | 0.86 (0.01) | 0.79 (0.02) | 0.81 (0.0) |
| SAHP+ | 0.29 (0.01) | 0.81 (0.0) | 0.88 (0.0) | 0.9 (0.0) | **0.83 (0.0)** |
| SAHP++ | **0.42 (0.01)** | **0.82 (0.0)** | 0.89 (0.0) | 0.9 (0.0) | **0.83 (0.0)** |
| LNM | 0.38 (0.02) | 0.76 (0.0) | 0.88 (0.0) | 0.89 (0.0) | 0.82 (0.0) |
| LNM+ | 0.41 (0.01) | 0.8 (0.0) | **0.89 (0.0)** | 0.9 (0.0) | **0.83 (0.0)** |
| LNM++ | **0.44 (0.01)** | 0.81 (0.0) | 0.89 (0.0) | 0.9 (0.0) | **0.83 (0.0)** |
| FNN | 0.3 (0.01) | 0.79 (0.0) | 0.89 (0.0) | 0.89 (0.0) | 0.81 (0.0) |
| FNN+ | 0.4 (0.01) | 0.81 (0.0) | 0.89 (0.0) | 0.9 (0.0) | **0.83 (0.0)** |
| FNN++ | **0.42 (0.01)** | **0.82 (0.0)** | 0.9 (0.0) | **0.91 (0.0)** | **0.83 (0.0)** |
| RMTPP | 0.39 (0.01) | 0.76 (0.0) | 0.86 (0.0) | **0.9 (0.0)** | 0.82 (0.0) |
| RMTPP+ | 0.39 (0.01) | 0.8 (0.0) | 0.87 (0.01) | **0.9 (0.0)** | **0.83 (0.0)** |
| RMTPP++ | **0.42 (0.01)** | **0.82 (0.0)** | 0.89 (0.0) | **0.9 (0.0)** | **0.83 (0.0)** |

Table 9: Accuracy@5 and MRR results of the different setups across all datasets. The values are computed over 5 splits, and the standard error is reported in parenthesis. Best results are highlighted in bold.

| | Accuracy@5 | | | | |
| | LastFM | MOOC | Github | Reddit | Stack O. |
|---|---|---|---|---|---|
| THP | 0.46 (0.01) | 0.87 (0.0) | 0.95 (0.01) | **0.93 (0.0)** | **0.93 (0.0)** |
| THP+ | 0.47 (0.01) | 0.89 (0.0) | 0.96 (0.0) | 0.92 (0.0) | **0.93 (0.0)** |
| THP++ | **0.52 (0.01)** | 0.9 (0.0) | 0.97 (0.0) | **0.93 (0.0)** | **0.93 (0.0)** |
| STHP+ | **0.46 (0.01)** | 0.88 (0.0) | 0.96 (0.0) | **0.92 (0.0)** | **0.93 (0.0)** |
| STHP++ | **0.46 (0.01)** | 0.89 (0.0) | 0.97 (0.0) | 0.91 (0.0) | 0.92 (0.0) |
| SAHP | 0.22 (0.01) | 0.67 (0.01) | 0.95 (0.0) | 0.84 (0.02) | 0.91 (0.0) |
| SAHP+ | 0.39 (0.01) | 0.89 (0.0) | **0.97 (0.0)** | 0.92 (0.0) | **0.93 (0.0)** |
| SAHP++ | **0.52 (0.01)** | 0.9 (0.0) | 0.97 (0.0) | **0.93 (0.0)** | **0.93 (0.0)** |
| LNM | 0.47 (0.02) | 0.88 (0.0) | 0.96 (0.0) | 0.92 (0.0) | 0.92 (0.0) |
| LNM+ | 0.51 (0.01) | 0.89 (0.0) | 0.97 (0.0) | **0.93 (0.0)** | **0.93 (0.0)** |
| LNM++ | **0.54 (0.01)** | 0.9 (0.0) | 0.97 (0.0) | **0.93 (0.0)** | **0.93 (0.0)** |
| FNN | 0.4 (0.01) | 0.88 (0.0) | **0.97 (0.0)** | 0.92 (0.0) | 0.91 (0.0) |
| FNN+ | 0.5 (0.01) | 0.9 (0.0) | 0.97 (0.0) | **0.93 (0.0)** | **0.93 (0.0)** |
| FNN++ | **0.53 (0.01)** | 0.9 (0.0) | 0.97 (0.0) | **0.93 (0.0)** | **0.93 (0.0)** |
| RMTPP | 0.48 (0.01) | 0.88 (0.0) | 0.96 (0.0) | **0.93 (0.0)** | 0.92 (0.0) |
| RMTPP+ | 0.49 (0.01) | 0.89 (0.0) | 0.96 (0.0) | **0.93 (0.0)** | **0.93 (0.0)** |
| RMTPP++ | **0.52 (0.01)** | 0.9 (0.0) | 0.97 (0.0) | **0.93 (0.0)** | **0.93 (0.0)** |

| | MRR | | | | |
| | LastFM | MOOC | Github | Reddit | Stack O. |
|---|---|---|---|---|---|
| THP | 0.32 (0.01) | 0.6 (0.0) | 0.74 (0.01) | **0.87 (0.0)** | **0.67 (0.0)** |
| THP+ | 0.34 (0.01) | 0.68 (0.0) | 0.77 (0.01) | **0.87 (0.0)** | **0.67 (0.0)** |
| THP++ | **0.38 (0.01)** | 0.7 (0.0) | 0.79 (0.01) | **0.87 (0.0)** | **0.67 (0.0)** |
| STHP+ | 0.33 (0.01) | **0.64 (0.0)** | 0.77 (0.01) | **0.86 (0.0)** | **0.67 (0.0)** |
| STHP++ | 0.34 (0.01) | **0.64 (0.0)** | 0.78 (0.0) | **0.86 (0.0)** | **0.67 (0.0)** |
| SAHP | 0.16 (0.0) | 0.5 (0.01) | 0.74 (0.01) | 0.76 (0.02) | **0.67 (0.0)** |
| SAHP+ | 0.27 (0.01) | 0.69 (0.0) | 0.78 (0.01) | 0.86 (0.0) | **0.67 (0.0)** |
| SAHP++ | **0.38 (0.01)** | 0.7 (0.0) | 0.79 (0.01) | **0.87 (0.0)** | **0.67 (0.0)** |
| LNM | 0.35 (0.01) | 0.62 (0.0) | 0.77 (0.0) | 0.86 (0.0) | 0.66 (0.0) |
| LNM+ | 0.37 (0.01) | 0.68 (0.0) | **0.79 (0.0)** | **0.87 (0.0)** | **0.67 (0.0)** |
| LNM++ | **0.4 (0.01)** | 0.69 (0.0) | 0.79 (0.01) | **0.87 (0.0)** | **0.67 (0.0)** |
| FNN | 0.27 (0.01) | 0.67 (0.0) | 0.79 (0.01) | 0.85 (0.0) | 0.66 (0.0) |
| FNN+ | 0.36 (0.01) | 0.7 (0.0) | 0.79 (0.0) | **0.87 (0.0)** | **0.67 (0.0)** |
| FNN++ | **0.38 (0.01)** | 0.7 (0.0) | 0.8 (0.0) | **0.87 (0.0)** | **0.67 (0.0)** |
| RMTPP | 0.36 (0.01) | 0.61 (0.0) | 0.75 (0.01) | **0.87 (0.0)** | 0.66 (0.0) |
| RMTPP+ | 0.36 (0.01) | 0.69 (0.0) | 0.77 (0.01) | **0.87 (0.0)** | **0.67 (0.0)** |
| RMTPP++ | **0.38 (0.01)** | 0.7 (0.0) | 0.79 (0.01) | **0.87 (0.0)** | **0.67 (0.0)** |

Table 10: MAE results of the different setups across all datasets. The values are computed over 5 splits, and the standard error is reported in parenthesis. Best results are highlighted in bold.

| | LastFM | MOOC | MAE Github | Reddit | Stack O. |
|---|---|---|---|---|---|
| THP | **0.01624 (0.00056)** | **0.13332 (0.00096)** | 0.0935 (0.01328) | **0.15257 (0.00322)** | **0.09885 (0.00097)** |
| THP+ | 0.01925 (0.00165) | 0.13638 (0.0016) | 0.09769 (0.014) | 0.15811 (0.00079) | 0.09972 (0.00077) |
| THP++ | 0.01744 (0.00103) | 0.13365 (0.00134) | **0.09327 (0.01162)** | 0.17627 (0.00359) | 0.09926 (0.00069) |
| STHP+ | **0.03717 (0.00978)** | 0.17422 (0.01899) | 0.11691 (0.00889) | **0.15864 (0.00409)** | **0.09883 (0.0007)** |
| STHP++ | 0.08356 (0.01887) | **0.17057 (0.02083)** | **0.1029 (0.01195)** | 0.22542 (0.02801) | 0.09931 (0.00056) |
| SAHP | 0.0137 (0.00055) | 0.12319 (0.00174) | 0.08409 (0.01092) | 0.18839 (0.00389) | 0.0992 (0.00066) |
| SAHP+ | 0.01364 (0.00056) | 0.12318 (0.00191) | 0.07491 (0.0117) | 0.13427 (0.00216) | **0.09853 (0.00071)** |
| SAHP++ | **0.01358 (0.0006)** | **0.12138 (0.00197)** | **0.07127 (0.0102)** | **0.13168 (0.00206)** | 0.10285 (0.00057) |
| FNN | 0.03563 (0.0138) | 0.16602 (0.0487) | 0.41847 (0.19173) | 0.23207 (0.05185) | 0.266 (0.09399) |
| FNN+ | 0.01055 (0.00043) | 0.07732 (0.00034) | **0.07093 (0.00902)** | 0.13157 (0.00208) | **0.09299 (0.00072)** |
| FNN++ | **0.01054 (0.00042)** | **0.07711 (0.00039)** | 0.07102 (0.0093) | **0.13148 (0.00231)** | 0.09305 (0.00064) |
| LNM | 0.0111 (0.00049) | 0.13516 (0.00078) | 0.08706 (0.01153) | **0.13494 (0.00202)** | 0.09494 (0.00075) |
| LNM+ | 0.01131 (0.00054) | 0.13548 (0.0011) | 0.08715 (0.0114) | 0.135 (0.00213) | 0.09517 (0.00069) |
| LNM++ | **0.01098 (0.00049)** | **0.13083 (0.00096)** | **0.08694 (0.01135)** | 0.1361 (0.00209) | **0.09476 (0.00071)** |
| RMTPP | **0.01619 (0.00097)** | 0.1208 (0.00084) | 0.12939 (0.02772) | **0.13577 (0.00223)** | **0.09522 (0.00069)** |
| RMTPP+ | 0.01677 (0.00072) | 0.12059 (0.00083) | 0.11416 (0.02624) | 0.1362 (0.00227) | 0.09563 (0.00085) |
| RMTPP++ | 0.01705 (0.00058) | **0.12011 (0.00079)** | **0.09183 (0.0106)** | 0.13772 (0.00192) | 0.09656 (0.0008) |

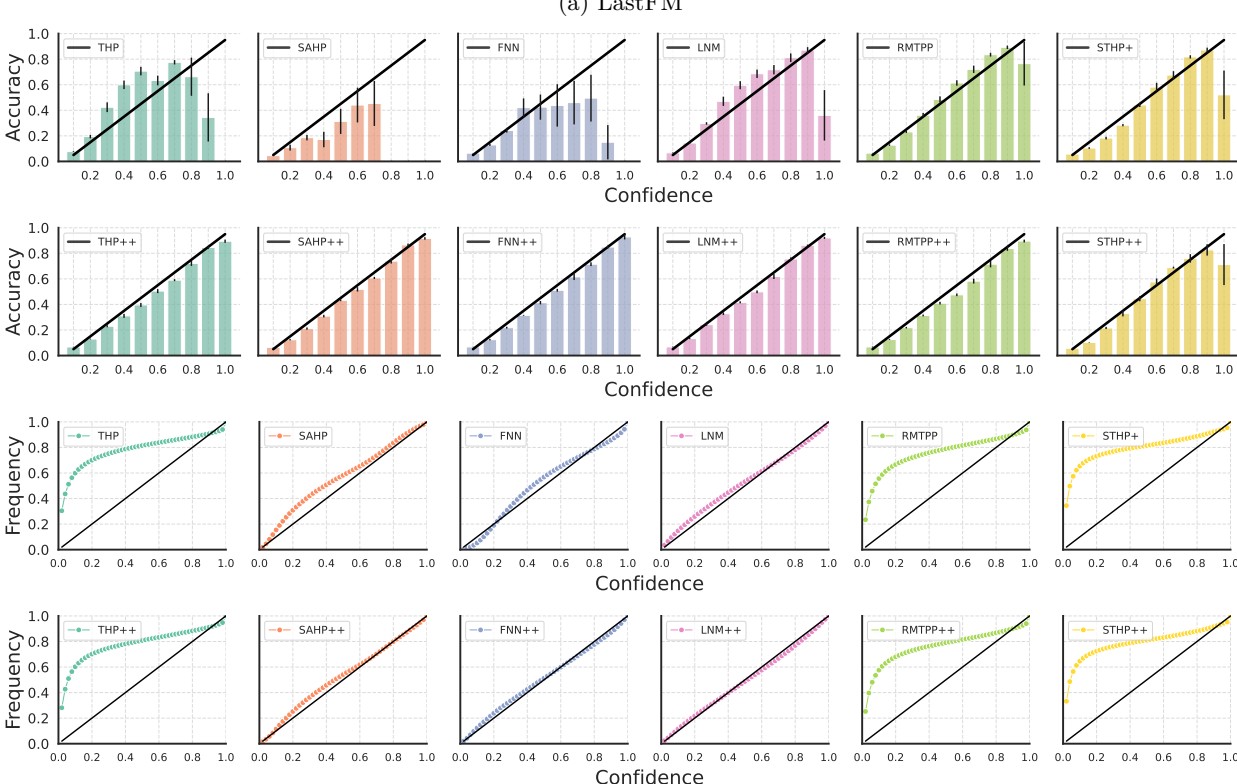

(a) LastFM

Figure 8: Reliability diagrams of the predictive $p^*(k|\tau; \boldsymbol{\theta}_M)$ (top) and $f^*(\tau, \boldsymbol{\theta}_T)$ (bottom) on LastFM. Frequency and Accuracy aligning with the black diagonal corresponds to perfect calibration. The results are averaged over 5 splits, and the error bars correspond to the standard error.

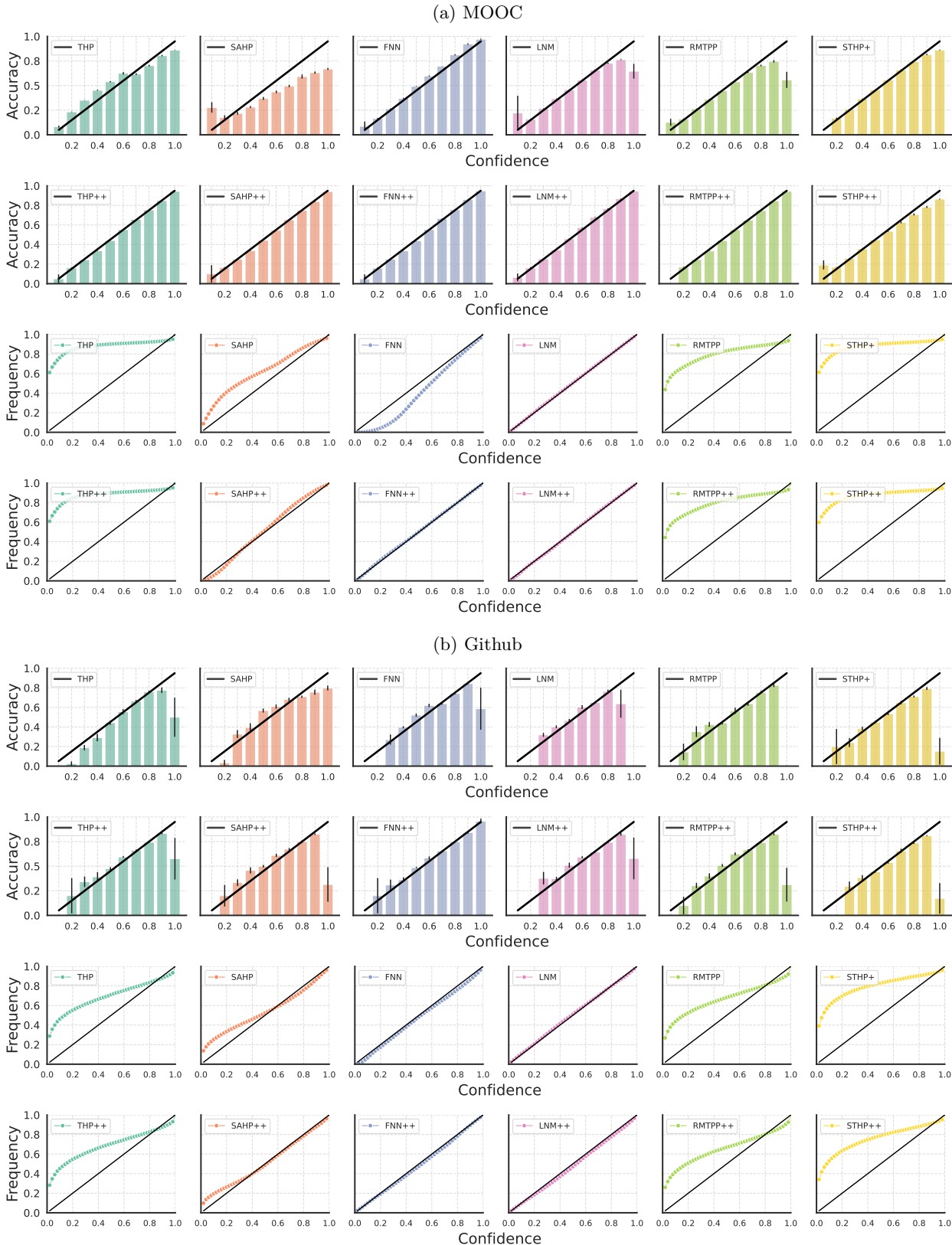

Figure 9: Reliability diagrams of the predictive $p^*(k|\tau; \boldsymbol{\theta}_M)$ (top) and $f^*(\tau, \boldsymbol{\theta}_T)$ (bottom) on MOOC anf Github. Frequency and Accuracy aligning with the black diagonal corresponds to perfect calibration. The results are averaged over 3 splits, and the error bars correspond to the standard error.

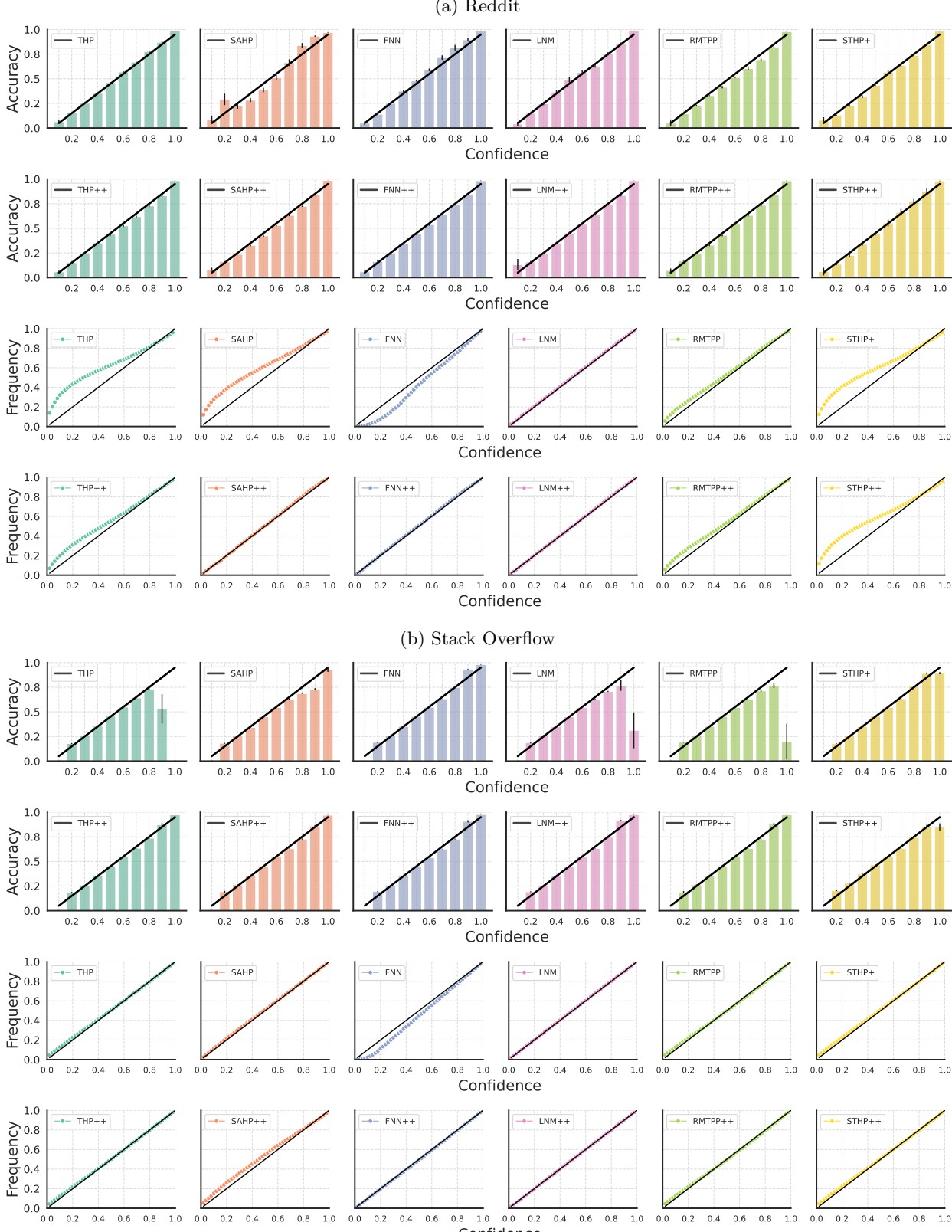

Figure 10: Reliability diagrams of the predictive $p^*(k|\tau; \boldsymbol{\theta}_M)$ (top) and $f^*(\tau, \boldsymbol{\theta}_T)$ (bottom) on Reddit and Stack Overflow. Frequency and Accuracy aligning with the black diagonal corresponds to perfect calibration. The results are averaged over 5 splits, and the error bars correspond to the standard error.

### H.3 Conflicting gradients remain harmful as capacity increases

To assess whether conflicting gradients remain detrimental to predictive performance for higher-capacity models, we train THP, SAHP and FNN in the base and base-DD settings while progressively increasing the number of trainable parameters. Figure 11 shows the evolution of the proportion of CG, GMS and TPI during training for these models on LastFM and MOOC, along with the evolution of their test $\mathcal{L}_T$ and $\mathcal{L}_M$ as a function of number of parameters. For each capacity (25K, 50K, 75K and 100K parameters), we maintain the distribution of parameters between the encoder and decoder heads constant at 0.67/0.33. Note that we only report the CG, GMS, and TPI values for the base models, as the base-DD setting is by definition free of conflicts. We note that increasing a model's capacity has a limited impact on the CG, GMS and TPI values, as well as on model performance with respect to both $\mathcal{L}_T$ and $\mathcal{L}_M$. In contrast, differences in performance between the base and base-DD setups are more significant, suggesting that conflicting gradients remain harmful to performance even with increased model capacity.

### H.4 Scaling the loss does not efficiently address conflicts

Our findings in Figure 4 suggest that conflicting gradients generally tend to favor $\mathcal{T}_T$ at the encoder heads during optimization, as illustrated by TPI values $> 0.5$. To better balance tasks during training, a natural approach would consist in scaling the contribution of $\mathcal{T}_T$ in (2) to reduce its impact on the overall loss, i.e.

$$\mathcal{L}(\theta; \mathcal{S}_{\text{train}}) = \frac{1}{s} \mathcal{L}_T(\theta; \mathcal{S}_{\text{train}}) + \mathcal{L}_M(\theta; \mathcal{S}_{\text{train}}), \tag{32}$$

where $s \geq 1$ is a scaling coefficient. To assess the effectiveness of this method, we train the base THP, SAHP and FNN models on the objective in (32) following the experimental setup detailed in Section E. For these models, we report in Figure (12) the evolution of the training CG, GMS and TPI values, along with their unscaled test $\mathcal{L}_T$ and $\mathcal{L}_M$. We observe that the occurrence of conflicting gradients is marginally impacted by larger values of $s$. However, as scaling increases, the magnitude of $\boldsymbol{g}_T$ diminishes, which translates into decreased values of GMS and TPI. Consequently, optimization begins to favor $\mathcal{T}_M$, and improvements on $\mathcal{L}_M$ can be observed. Nevertheless, the crashed TPI and GMS values are at the root of significant degradation with respect to $\mathcal{L}_T$, offsetting the gains on $\mathcal{T}_M$. Although a specific value of $s$ could lead to a trade-off between tasks, models trained in our base+ or base++ settings generally show improved performance with respect to both tasks simultaneously, as shown in Table 1.

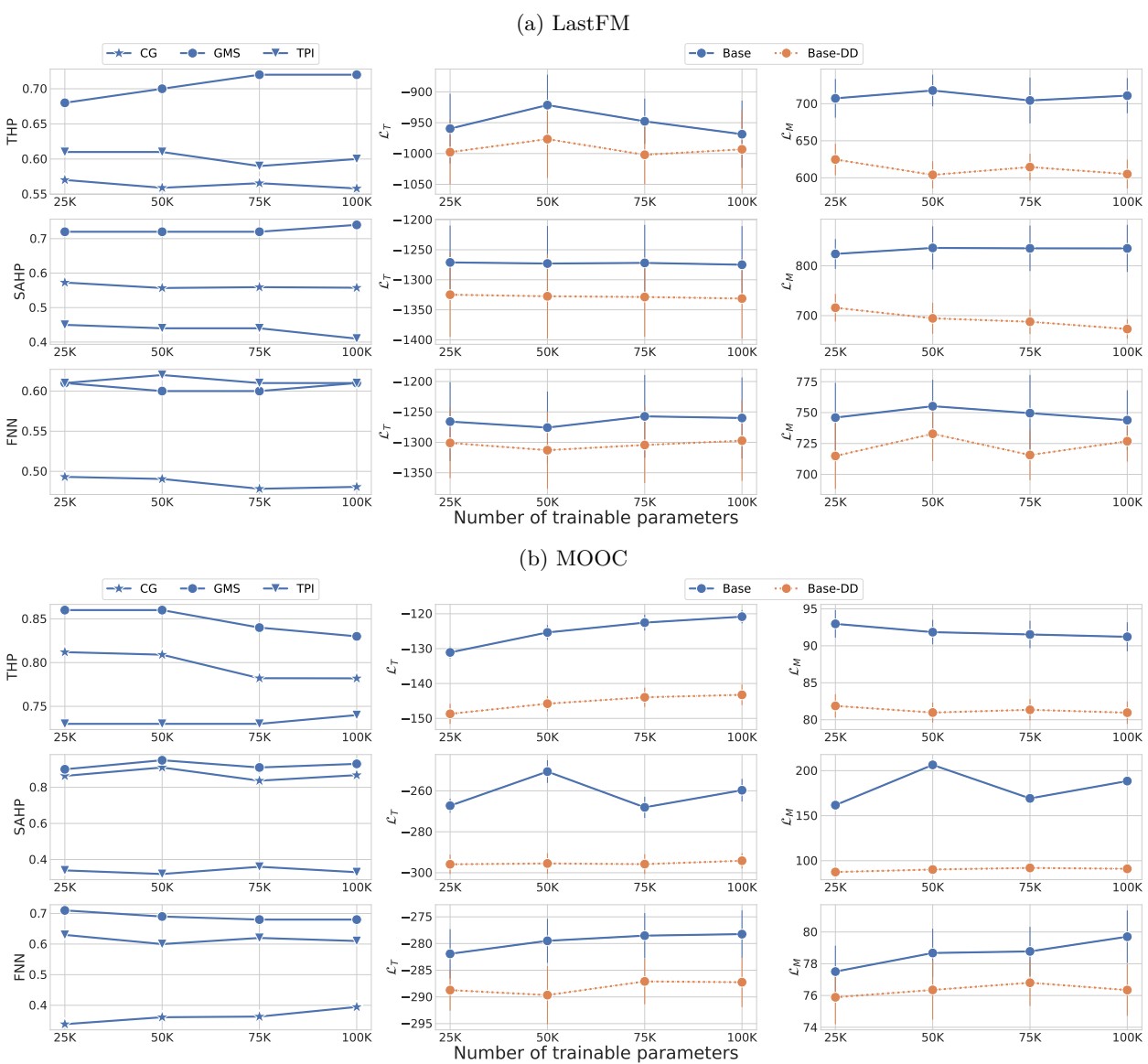

Figure 11: Left: Evolution of CG, GMS, TPI with increasing model capacity for the base THP, SAHP and FNN during training on LastFM and MOOC. Right: Evolution of the test $\mathcal{L}_T$ and $\mathcal{L}_M$ in the base and base-DD settings.

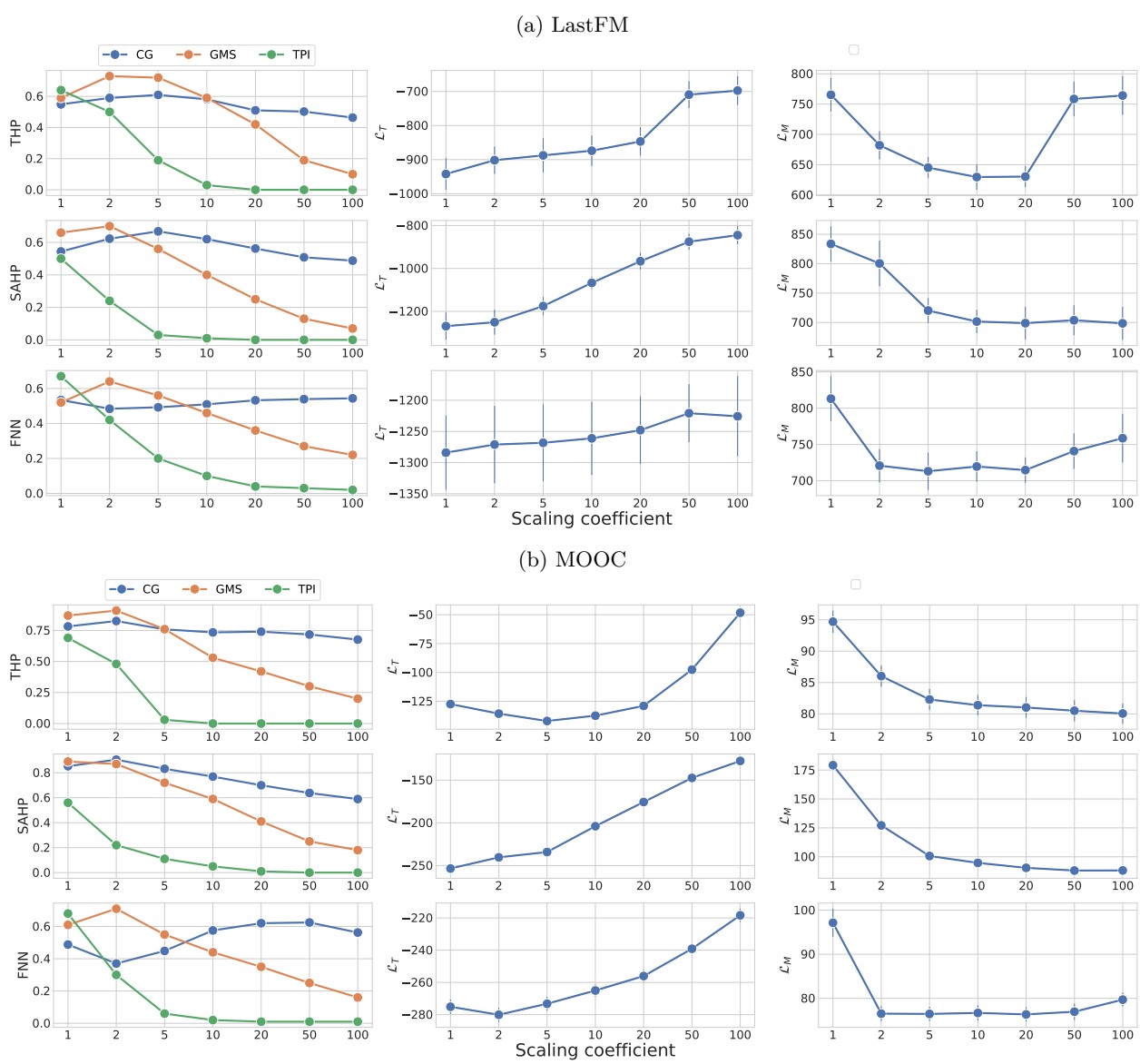

Figure 12: Left: Evolution of CG, GMS, TPI with scaling coefficient $s$ for the base THP, SAHP and FNN during training on LastFM. Right: Evolution of the test $\mathcal{L}_T$ and $\mathcal{L}_M$ in the base and base-DD settings.

## H.5 Distributions of conflicting gradients between the base and base+ settings

Figures (13) to (14) show the distributions of cos $\phi_{TM}$ during training, as well as the proportion of conflicting gradients (CG), their average GMS, and their average TPI for all models in the base and base+ setups on all datasets. As discussed in the main text, severe conflicts (cos $\phi_{TM} < -0.5$) often arise when training neural MTPP models in the base setup. For THP, SAHP, and FNN, the base+ setup reduces the occurrence of severe conflicts at the encoder heads, and prevents conflicting gradients to appear at the decoder heads. For LNM and RMTPP, we only show the distribution of cos $\phi_{TM}$ at the encoder heads as the decoders are by design free of conflicting gradients in the base and base+ setups. For these models, the base+ setup does not change significantly the distribution of conflicting gradients during training, which aligns with expectations. Hence, for LNM+ and RMTPP+, we attribute performance gains on the mark prediction task to the relaxation of conditional independence between arrival times and marks via (9).

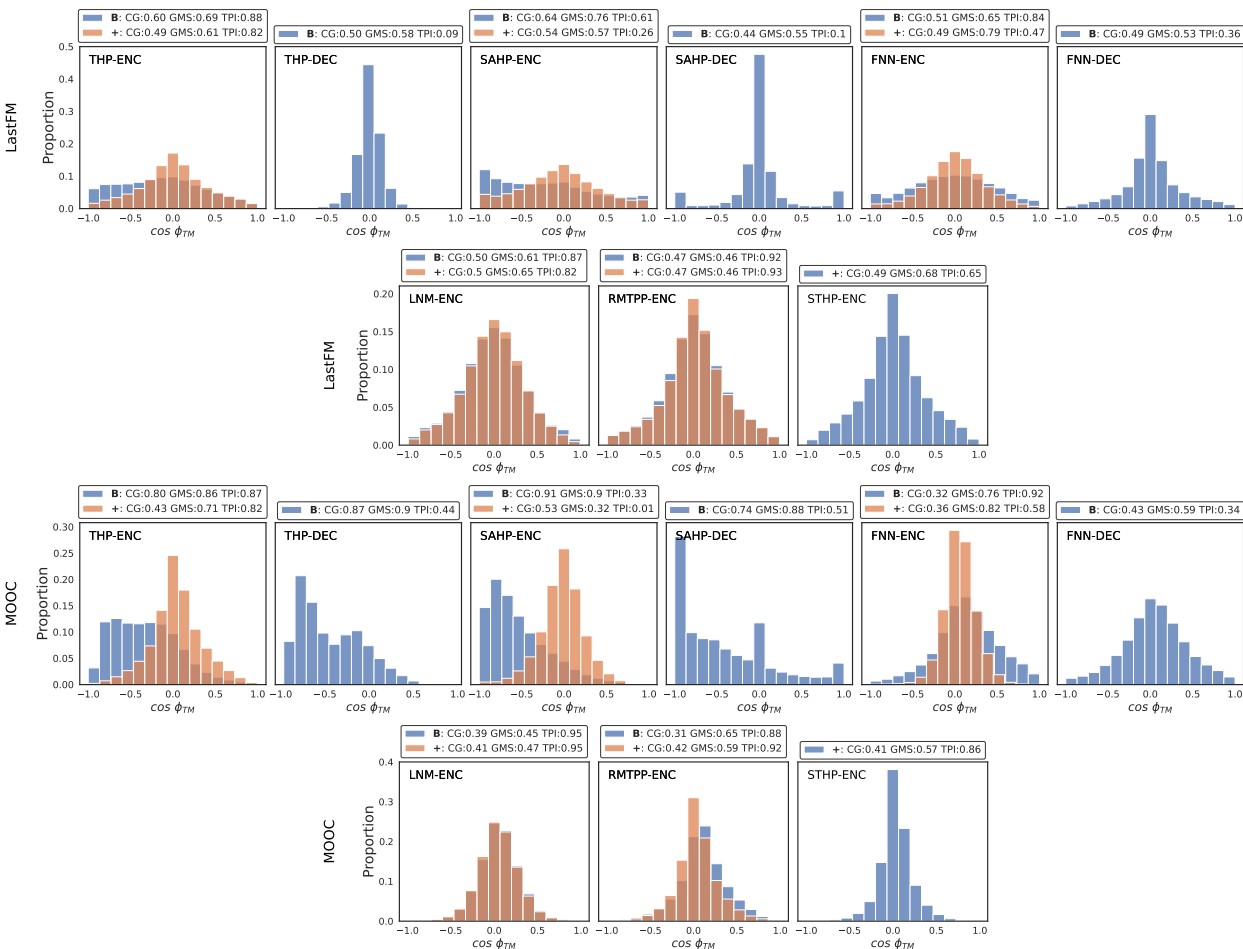

Figure 13: Distribution of cos $\phi_{TM}$ during training at the encoder (ENC) and decoder (DEC) heads for all baselines in the base and base+ setup on LastFM and MOOC. "B" and "+" refer to the base and base+ models, respectively, and the distribution is obtained by pooling the values of $\phi_{TM}$ over 5 training runs. As the decoders are disjoint in the base+ setting, note that cos $\phi_{TM}$ is not defined.

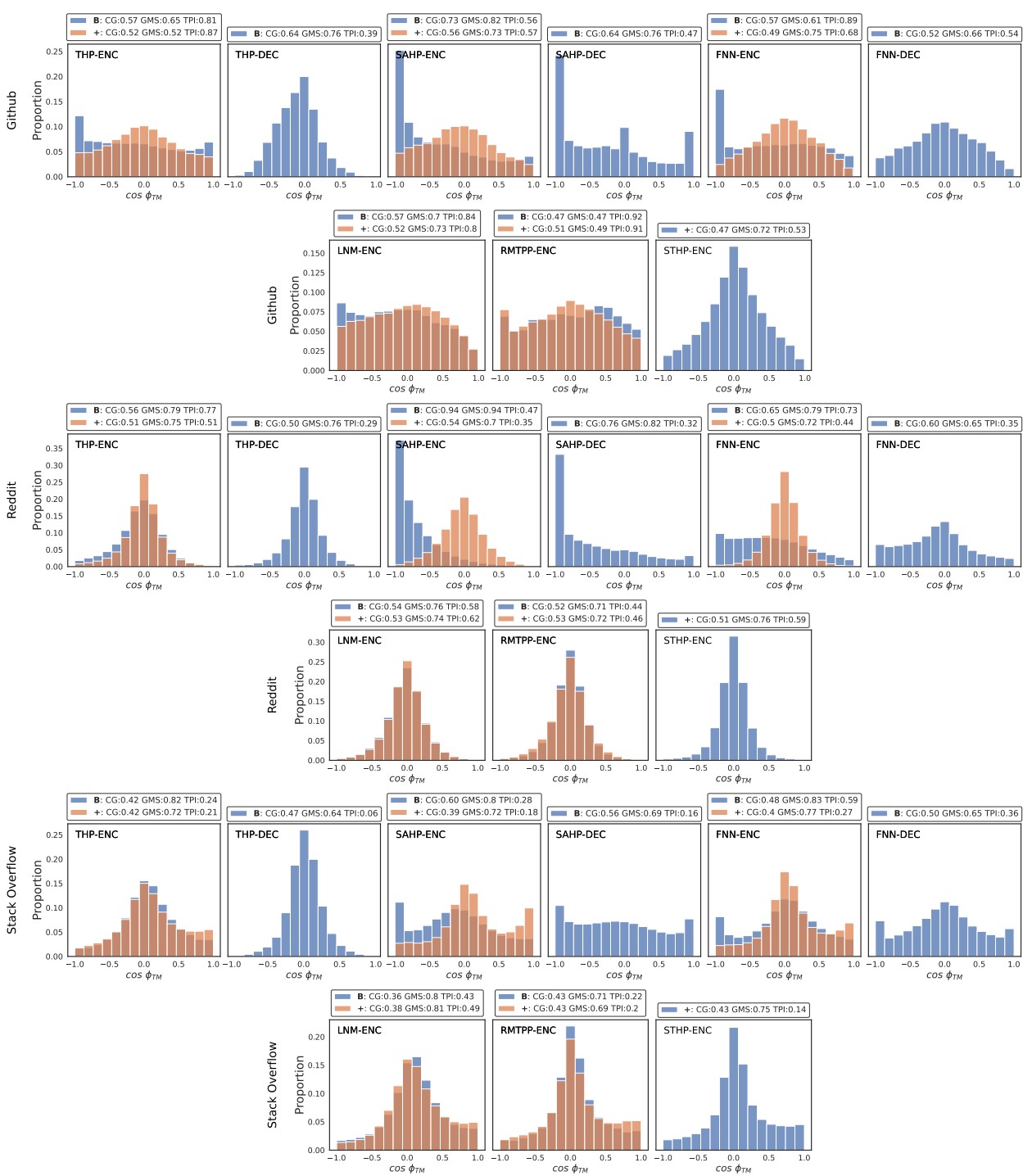

Figure 14: Distribution of cos $\phi_{TM}$ during training at the encoder (ENC) and decoder (DEC) heads for all baselines in the base and base+ setup on Github, Reddit, and Stack Overflow. "B" and "+" refer to the base and base+ models, respectively, and the distribution is obtained by pooling the values of $\phi_{TM}$ over 5 training runs. As the decoders are disjoint in the base+ setting, note that cos $\phi_{TM}$ is not defined.

## H.6 Distributions of conflicting gradients between the base and base-D settings

Figure (15) show the distribution of cos $\phi_{TM}$ during training, as well as the proportion of conflicting gradients (CG), their average GMS, and their average TPI for THP, SAHP, and FNN in the base and base-D setups on all datasets. We observe that using two identical and task-specific instances of the same decoder in the base-D setup for the time and mark prediction tasks mitigates the occurrence of conflicts. The decrease in conflicting gradients during training is in turn associated to increased performance with respect to both tasks. We refer the reader to the discussion in Section 6.1 of the main text for further details.

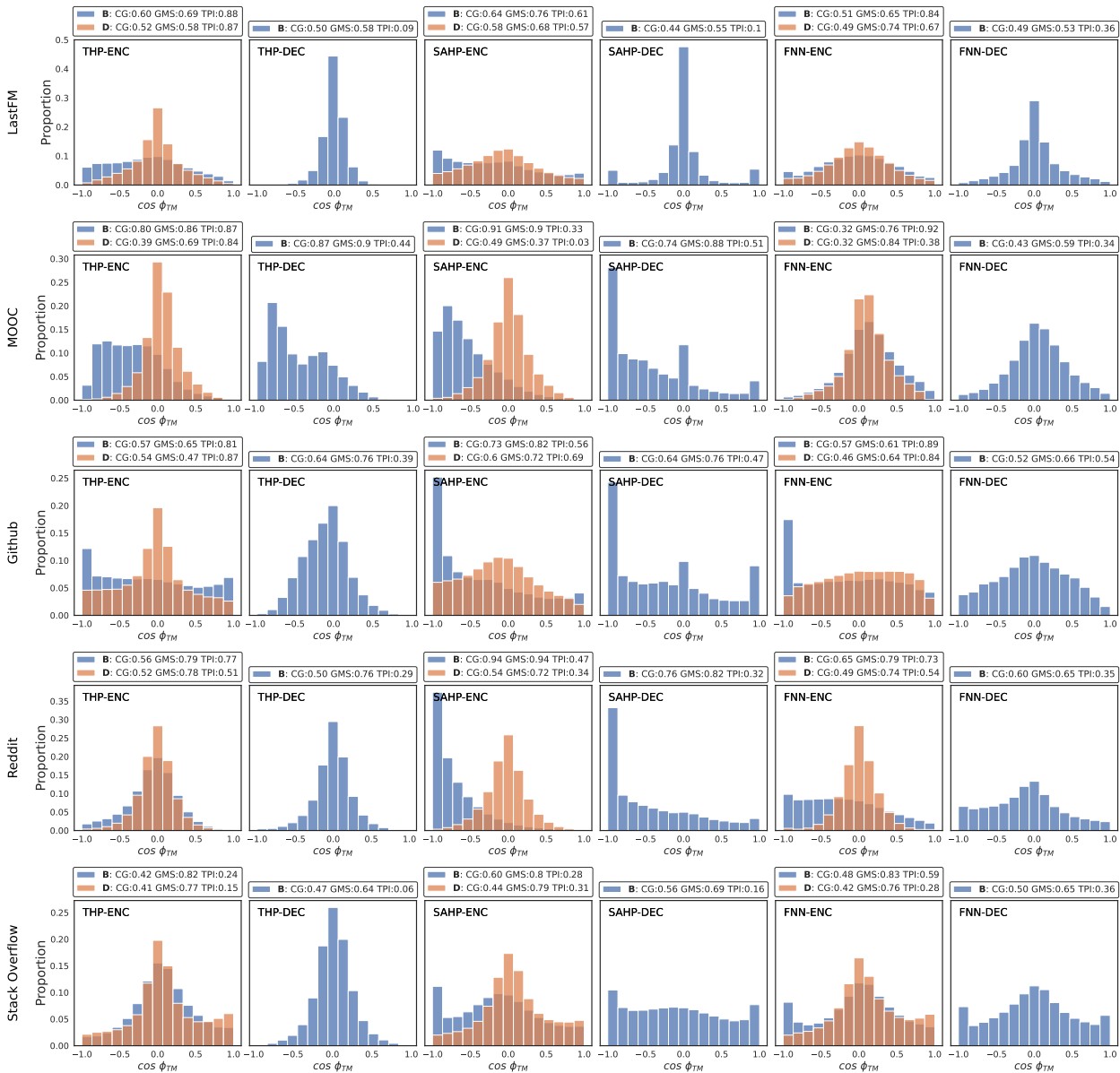

Figure 15: Distribution of cos $\phi_{TM}$ during training at the encoder (ENC) and decoder (DEC) heads for all baselines in the base and base-D setup on all datasets. "B" and "D" refer to the base and base-D models, respectively, and the distribution is obtained by pooling the values of $\phi_{TM}$ over 5 training runs. As the decoders are disjoint in the base-D setting, note that cos $\phi_{TM}$ is not defined.

