# OpenReview forum: "Preventing Conflicting Gradients in Neural Marked Temporal Point Processes"
_TMLR — Accepted by TMLR_

### Review · Reviewer_JXsD · 2024-10-14

**Summary Of Contributions:**

This work first investigated and validated that the inference in neural marked temporal point processes would be affected by the conflicting gradients issue in which the gradients from the temporal event modeling has opposite direction with the gradients from the mark events. To address this conflicting gradients issue, it proposed a framework in which it proposed disjoint history encoders/decoders.

**Audience:**

Yes

**Claims And Evidence:**

Yes

**Requested Changes:**

See Weakness

**Strengths And Weaknesses:**

Strengths:
1. It studies that conflicting gradients issue in the neural marked temporal point process, showing that this issue existed in multiple public models such as THP, SAHP and etc.
2. It proposed a novel framework to address the conflicting gradient by disjointly modeling the conditional probability and intensity function in terms of history.

Weaknesses:
1. Need more discussion for the results. For the Table 5, since base model and base+ model has different parameters, comparing the cosine distance between gradients is not straight-forward. More discussion would be preferred.
2. Although Table 1 show the benefits from disjoint modeling. But that benefits may comes from the model sizes since disjoint modeling would double model parameters for history modeling. More experiments and discussions are needed.

---

> ### Author Response · Authors · 2024-11-05
> **Reply to reviewer JXsD**
>
> Thank you for your constructive feedback on our work. We have addressed your concerns below.
>
> **Clarification on conflicting gradients in Figures 5 and 7.**
>
> We believe that Reviewer JXsD was referring to the distribution of conflicting gradients in Figures 5 and 7, as Table 5 primarily presents computational trade-offs across different approaches. To clarify, while the decoder heads do differ between the base and base+ (or base-D) configurations, the encoder heads remain consistent across both settings (i.e., a GRU with approximately the same number of hidden units). Therefore, the computation of conflicting gradients remains consistent between base and base+ (base-D), allowing for a direct comparison in Figures 5 and 7. For the decoder heads, since the parameters are disjoint by definition in the base+ setting, we cannot assess the angle between their gradients. This clarification has been added to the revised version.
>
> **Controlling for the number of parameters.**
>
> We appreciate this valid concern, which we carefully accounted for in our experiments. As illustrated in Table 4, we ensured parameter counts were controlled across settings, keeping them approximately equal (both overall and between encoder and decoder heads). This approach allows us to isolate performance gains from simply increased model capacity.

---

> > ### Comment · Reviewer_JXsD · 2024-11-27
> > **Response**
> >
> > Thanks for the clarification and it has addressed my concerns.

---

### Review · Reviewer_fNFi · 2024-10-22

**Summary Of Contributions:**

It is shown that the training of Marked Temporal Point Process (MTTP) constitutes a two-task learning problem. During training, the gradients of the different tasks can counteract each other. In the paper, the authors propose different ways of parametrizing the model to alleviate this issue. Multiple experiments are conducted showing improved alignment between the gradients (where applicable) as well as slightly improved performance on other metrics.

**Audience:**

Yes

**Claims And Evidence:**

Yes

**Requested Changes:**

*) Definition 3:

The formulation is unclear:"s.t. $cos(\phi_{TM})<0$" seems out of place at its current position. For example,  "... conflicting gradients g_T and g_M (i.e. when $cos(\phi_{TM})<0$) ..." would be clearer.

*) Page 4: "...most models also show a certain proportion of positive gradients ($cos \phi_{TM} \geq 0$) ..."

"positive gradients" -> "non-conflicting gradients"

This is a bit of an understatement, in many of the models the proportion of positive and negative alignment are almost equal. In Section 6, it would be interesting to comment on whether a bigger improvement in terms of the evaluation metrics is observed for models that had a higher proportion of conflicting gradients in Fig. 2.

*) Figure 6:

(left) (right) -> (right)

**Strengths And Weaknesses:**

Strengths:

*) The paper is well written and the aim is well motivated.

*) The suggested methods appear to be effective at bypassing the negative impact of conflicting gradients.

Weaknesses:

*) 3 runs do not allow for accurate estimates of the standard error (e.g. in Table 1). The results would be strengthened by increasing the number of runs.

---

> ### Author Response · Authors · 2024-11-05
> **Reply to reviewer fNFi**
>
> Thank you for your insightful comments and valuable suggestions to improve our work. We have addressed your concerns in the following points.
>
> **Additional training runs.**
>
> In response to your recommendation, we increased the number of runs from 3 to 5 using two additional splits across all datasets, and we updated all Tables and Figures accordingly. We found no significant changes in the results.
>
> **Clarification of Definition 3.**
>
> Thank you for your suggestion. We have updated Definition 3 for clarity in the revised version.
>
> **Clarification regarding the proportion of conflicting gradients.**
>
> We have added clarifications to emphasize that conflicting gradients are detrimental to model performance when (1) they appear in high proportion during training and (2) they are associated with low GMS values. For example, in Figure 2, THP and SAHP show high CG levels with high GMS values, while the other models exhibit a more balanced CG but with lower GMS values. We observe that both scenarios lead to similar performance deterioration for both tasks, as most models show comparable improvements on $\mathcal{L}_T$ and $\mathcal{L}_M$ when trained in the base++ setting.
>
> **Typo in Figure 6.**
>
> Thank you for noting this. We have corrected the caption.

---

> > ### Comment · Reviewer_fNFi · 2024-11-20
> >
> > Thank you for the reply, my concerns have been addressed.

---

### Review · Reviewer_6Ygz · 2024-10-25

**Summary Of Contributions:**

The paper proposes to model MTPP as two multi-task learning problems, where one task focuses on learning the distribution of the next event's arrival time conditional on historical events, and the other learns the distribution of the categorical mark conditional on both the event's arrival time and the historical events. The paper argues that modeling these two jointly with common model parameters might result in a sub-optimal solution due to conflicting gradients between the two tasks. The conflicting gradient refers to having two gradients in opposite directions where one has a significantly larger magnitude than the other, so the learning process tends to favor tasks with larger gradient magnitudes.

The paper proposes to use the following measures: (a) conflicting gradient as the cos $\phi$ angle between the gradient of the two tasks, (b) gradient magnitude similarity (GMS) to measure the difference in magnitude of the two gradients; the paper argues that this only measures that there is a significant gradient difference to measure the direction of the difference (c) they propose the time priority index (TGI) between conflicting gradients to determine the direction of the conflict. Ideally, cos $\phi$>0, while GMS is around 1. They then performed experiments on a set of MTPP models and showed that they exhibit conflicting gradients.

# Approach:

The paper proposes to solve this problem by disjoint parametrizations of neural MTPP models they investigate:

- "Shared History Encoders and Disjoint Decoders," where a common history encoder encodes time and marks predictive history together, while the two predictive functions are modeled independently through separate decoders. They referred to this model as Base +.

- "Disjoint History Encoders and Disjoint Decoders." Each function has an independent history encoder and an independent decoder. They referred to this model as Base ++.

- "Shared History Encoders and Duplicated Decoders" instead of having two decoders trained independently, they have two identical parametrizations of the same base decoder. They referred to this model as Base -D

- "Disjoint History Encoders and Duplicated Decoder" is similar to Base ++, but instead, the decoders are duplicates of each other, as in Base -D. They refer to this as Base-DD

Here, base-D and base-DD settings retain the same architecture as the base model.

# Experiments

They evaluate these parameterization approaches on 6 different models and 5 different datasets.

## Results:

- Comparing regular training vs. Shared History Encoders and Disjoint Decoders (Base vs. base+), they reported that (a) generally conflicting gradient decrease, (b) GMS seems not to have changed much or often worsens, (c) decrease in both losses of mark and time.

- When shared History Encoders and Disjoint Decoders are compared to Disjoint History Encoders and Disjoint Decoders (base+ vs. base++), mark loss improves.

- Comparing regular training to Disjoint History Encoders and Decoders (Base vs. base++), it was found that the base++ models were better calibrated than the base models.

- When regular training is compared to Shared History Encoders and Duplicated Decoders (Base vs. base-D), the conflicting gradient decreases, and task accuracy improves.

**Audience:**

Yes

**Claims And Evidence:**

Yes

**Requested Changes:**

- Please explicitly mention how you get CG in the text from cos $\phi$.
- Figures 5 and 7  is missing y axes.
- Please comment on the GMS and TGI results in different experiments.

**Strengths And Weaknesses:**

# Strengths:
- The paper is very well written and the approach is well motivated.
- The proposed approach is relatively simple yet very effective across different models.
- Very thorough experimental section they investigated the usage of different parameterization schemes across  6 different models and 5 different datasets.
- Over all I think its a strong paper with strong empirical results.



# Weakness:
- Results on more recent models like AttNHP.
- GMS was introduced in the beginning of the paper as way to ensure that both gradients have the same magnitude the different parameterization schemes proposed in the paper did not help in this particular metric (since the have different gradients now I guess it doesn't matter since each model is separate) but it seemed confusing to introduce metric and not use it for comparison similarly to TGI.

---

> ### Author Response · Authors · 2024-11-05
> **Reply to reviewer 6Ygz**
>
> Thank you for your detailed review and positive feedback on our work. We address your concerns in the following paragraphs.
>
> **Results on more recent models, such as AttNHP.**
>
> We appreciate the reviewer’s suggestion regarding the inclusion of the AttNHP model. Our primary objective is to demonstrate the frequent emergence of conflicting gradients during the training of neural MTPP models and to introduce a simple re-parametrization framework that effectively prevents these occurrences. While we could have considered additional baselines, such as AttNHP, we chose to focus on six of the most widely recognized neural MTPP models in the literature. Furthermore, AttNHP shares similarities with THP and SAHP, two baselines already included in our analysis, which we believe adequately represent the model’s characteristics.
>
> **Obtaining CG from the gradients.**
>
> In Figures 2, 5, and 7, the proportion of conflicting gradients (CG) is computed as
> $$
> \text{CG} = \frac{\sum_{s=1}^S \mathbb{1}\left(\text{cos } \phi_{TM}^s < 0\right)}{\sum_{s=1}^S \text{cos } \phi_{TM}^s},
> $$
> where $\phi^s_{TM}$ refers to the angle between $\boldsymbol{g}_T^s$ and $\boldsymbol{g}_M^s$ at training iteration $s$. This has been explicitly clarified in the revised version.
>
> **Figures 5 and 7 are missing axes.**
>
> Thank you for bringing this to our attention. We have added the missing axes in the revised version.
>
> **More discussion on GMS and TPI.**
>
> Thank you for highlighting the need for additional discussion on GMS and TPI. While we provided a more in-depth analysis of the GMS and TPI metrics in Appendices H.3 and H.4, we agree that these metrics warrant further discussion in the main text.
>
> Notably, while GMS values do not consistently improve between the base and base+ settings in Figure 5, Table 1 shows improvements in both $\mathcal{L}_T$ and $\mathcal{L}_M$ across most models. This pattern suggests that the observed increase in GMS is often counterbalanced by fewer training conflicts, ultimately enhancing model performance on both tasks. Furthermore, TPI values indicate that the base+ configuration generally achieves a more balanced training outcome for both tasks, contributing to their performance gains.
>
> We have included this discussion in Section 6 of the revised manuscript.

---

> > ### Comment · Reviewer_6Ygz · 2024-11-24
> > **Thank you**
> >
> > Thank you for the clarification my concerns have been addressed.

---

### Author Response · Authors · 2024-11-05
**We thank the reviewers for their feedback**

We would like to thank all reviewers for constructive and helpful feedback that improved the quality of the paper. We have uploaded the revised version of the manuscript, highlighting changes from the original submission in blue.

---

### Author Response · Authors · 2025-01-14
**Camera-ready revision**

We thank the reviewers and the action editor for their constructive feedback during the reveiwing process, which contributed to enhancing the quality of our paper. We have now updated the camera-ready version, addressing the suggestions provided. Additionally, we have included a link to our code base within the paper.

---

### Decision · Action_Editor_7gR6 · 2024-12-06

**Recommendation:** Accept as is

**Comment:**

The authors have successfully addressed the reviewer feedback, which has strengthened the submission. All reviewers are in agreement that after modifications, the submission is ready for acceptance, which I agree with.

**Audience:**

This paper investigates a topic of interest to some in the TMLR audience.

**Claims And Evidence:**

This work explores the issue of conflicting gradients arising when framing Neural Marked Temporal Point Processes as a two-task optimization problem. They propose a disjoint parameterization of the model(s) to effectively avoid having conflicting gradients, evaluated on a series of real-world datasets.

The claims are well supported by the empirical evidence, especially after the suggestions provided by the reviewers.